**communications** engineering

# Solutions for decarbonising urban bus transport: a life cycle case study in Saudi Arabia
Chengcheng Zhao [1], Leiliang Zheng Kobayashi[1], Awad Bin Saud Alquaity[2,3], Jean-Christophe Monfort [4], Emre Cenker[4], Noliner Miralles[5] & S. Mani Sarathy [1] ✉

With heavy reliance on fossil fuels, countries like Saudi Arabia face challenges in reducing carbon emissions from urban bus transportation. Herein, we address the gaps in evaluating proton-exchange membrane fuel cell buses and develop a globally relevant life-cycle assessment model using Saudi Arabia as a case study. We consider various bus propulsion technologies, including fuel cell buses powered by grey and blue hydrogen, battery electric buses, and diesel engines, and include the shipping phase, air conditioning load, and refuelling infrastructure. The assessment illustrates fuel cell buses using blue hydrogen can reduce emissions by 53.6% compared to diesel buses, despite a 19.5% increase in energy use from carbon capture and storage systems. Battery electric buses are affected by the energy mix and battery manufacturing, so only cut emissions by 16.9%. Sensitivity analysis shows climate benefits depend on energy sources and efficiencies of carbon capture and hydrogen production. By 2030, grey and blue hydrogen-powered fuel cell buses and battery electric buses are projected to reduce carbon emissions by 19.3%, 33.4%, and 51% respectively, compared to their 2022 levels. Fully renewable-powered battery electric buses potentially achieve up to 89.6% reduction. However, fuel cell buses consistently exhibit lower environmental burdens compared to battery electric buses.

Greenhouse gas (GHG) reduction in non-member countries of the Organisation for Economic Co-operation and Development (OECD) is of paramount importance due to annual GHG growth rates being nine times higher than in OECD countries and the slower assimilation of decarbonisation technologies in all sectors[1-3]. To explore the decarbonisation prospects of non-OECD countries, we conducted a case study focusing on Saudi Arabia as a representative of mobility and power in non-OECD countries. Saudi Arabia has pledged to achieve net zero by 2060, with a mid-term target announced in Vision 2030[4]. GHG emissions stemming from transportation in Saudi Arabia account for 22%[5,6]. In line with Vision 2030 and the Sustainable Development Goals, Saudi Arabia confronts the urgent task of diminishing dependence on private vehicles while enhancing urban bus transportation. Integrating innovative vehicle technologies to decarbonise the public transport sector is of paramount importance for Saudi Arabia.

Hydrogen fuel cell vehicles (FCVs) and battery electric vehicles (BEVs) are emission-free in the vehicle operation phase, offering a pathway to decarbonise the urban bus transport[7-23]. However, the life-cycle environmental impacts of these vehicles are intrinsically linked to many other factors such as feedstock production, energy sources for electricity generation, fuel production efficiency, battery manufacturing, fuel cell (FC) stack performance, battery technology, and operational energy efficiency[7-9,11,13,17,19-25]. An extensive life-cycle analysis (LCA) is imperative for a comprehensive understanding of FCVs' and BEVs' decarbonisation potential.

Different colours are used to classify hydrogen production methods and energy sources[26]. Green hydrogen arises from water electrolysis using renewable energy[26]. Blue and grey hydrogen originate from steam methane reforming using natural gas (NG), differentiated by the employment of carbon capture and storage (CCS) for blue hydrogen[26]. Currently, the latter

[1]CCRC, Physical Sciences and Engineering (PSE) Divison, King Abdullah University of Science and Technology (KAUST), Thuwal 23955, Saudi Arabia. [2]Department of Mechanical Engineering, King Fahd University of Petroleum & Minerals (KFUPM), Dhahran 31261, Saudi Arabia. [3]Center for Hydrogen and Energy Storage, King Fahd University of Petroleum & Minerals (KFUPM), Dhahran 31261, Saudi Arabia. [4]Transport Technologies Division, R&DC, Saudi Aramco, Dhahran 31311, Saudi Arabia. [5]Saudi Public Transport Company (SAPTCO), Riyadh 11443, Saudi Arabia. ✉e-mail: mani.sarathy@kaust.edu.sa

two offer cost-effective, technically feasible, and commercially accessible supply relative to green hydrogen[27–29]. In 2021 only 4% of global hydrogen production exploited renewable energy, as opposed to 48% from NG[28]. Saudi Arabia has substantial NG reserves and also plans to further increase its production capacity[30], which makes grey and blue hydrogen more accessible and feasible currently and in the near future.

Furthermore, despite the escalating importance of FCVs and BEVs for road transport, the literature appears to be strikingly sparse, particularly for urban buses[31]. This research gap may largely be attributed to data scarcity of material inventories for these vehicles. The analyses pertaining to proton-exchange membrane (PEM) fuel cell buses (FCBs) are virtually absent, and those for battery electric buses (BEBs) are notably sparse. Existing research typically narrows its scope to powertrain systems[14]. Several LCA studies have identified a notable research gap in relation to evaluating the infra-structure construction associated with electric charging stations and hydrogen refuelling stations (RFSs)[31]. Furthermore, there is an absence of research on FCVs and BEVs in the Saudi Arabian context.

To bridge these research gaps and explore the decarbonisation potential of advanced vehicle technologies, this study provides an ana-lysis of emissions and energy utilisation of FCBs, BEBs, and internal combustion engine buses (ICEBs) across their entire life cycle, including the energy and emissions related to bus transportation via roll-on/roll-off vessels and the construction of charging and hydrogen RFS infra-structures. This was achieved by the case study of a potential bus fleet operating in Makkah, Saudi Arabia, with scenarios for 2022 and 2030. Achieving 100% renewable electricity for BEBs and transitioning to green hydrogen for FCBs present challenges due to uncertainties in renewable electricity implementation and green hydrogen feasibility in the short-to-medium term[4]. This study evaluates practical dec-arbonisation strategies involving FCBs powered by blue and grey hydrogen, leveraging NG resources and carbon capture infrastructure, as well as grid-connected BEBs, as practical transitional decarbonisation

solutions[32]. Additionally, hypothetical scenarios involving fully renew-able BEBs and FCBs powered by a projected 2030 hydrogen mix are explored to thoroughly assess their decarbonisation potential. Sensitivity analyses are performed, along with evaluations of key environmental impact indicators. Furthermore, the study introduces a globally applicable LCA model for bus systems, predicated on non-OECD con-ditions, yet universally adaptable. This model expands the GREET fra-mework to account for energy consumption and emissions across the bus's entire life-cycle, and bus air conditioning (AC) system energy demands.

## Results

### LCA goal and scope

This study explored the complete life-cycle of a city bus fleet operating in Makkah, Saudi Arabia, including both the fuel cycle and the vehicle cycle (Fig. 1). Furthermore, the RFS infrastructure construction was also resear-ched, and results were added to the fuel cycle. The bus route was decided based on real cases, and three bus models—Toyota Sora, BYD K9, and Volvo B8RLE—were chosen to represent FCB, BEB, and ICEB options, respectively.

In an urban bus transportation framework, the study focuses on a fleet comprising 20 buses—18 for regular operation and 2 as reserves. Each bus is expected to operate for 10 years, travelling a total of 508,080 kilometres. The study's scope encompasses a specific urban trajectory from Shimeisy Police Station to Al Haram in Makkah, with empirical data obtained through collaboration with the bus company. The fleet adheres to a structured schedule, where each bus completes two rounds daily. Operational design involves a 2-hour active period for each round, interspersed with a 3-hour rest period, thereby optimising fleet performance. Service frequency is set at 20-minute intervals, spanning from 08:00 to 22:00, to cater to the year-round Umrah-related demand peaks.

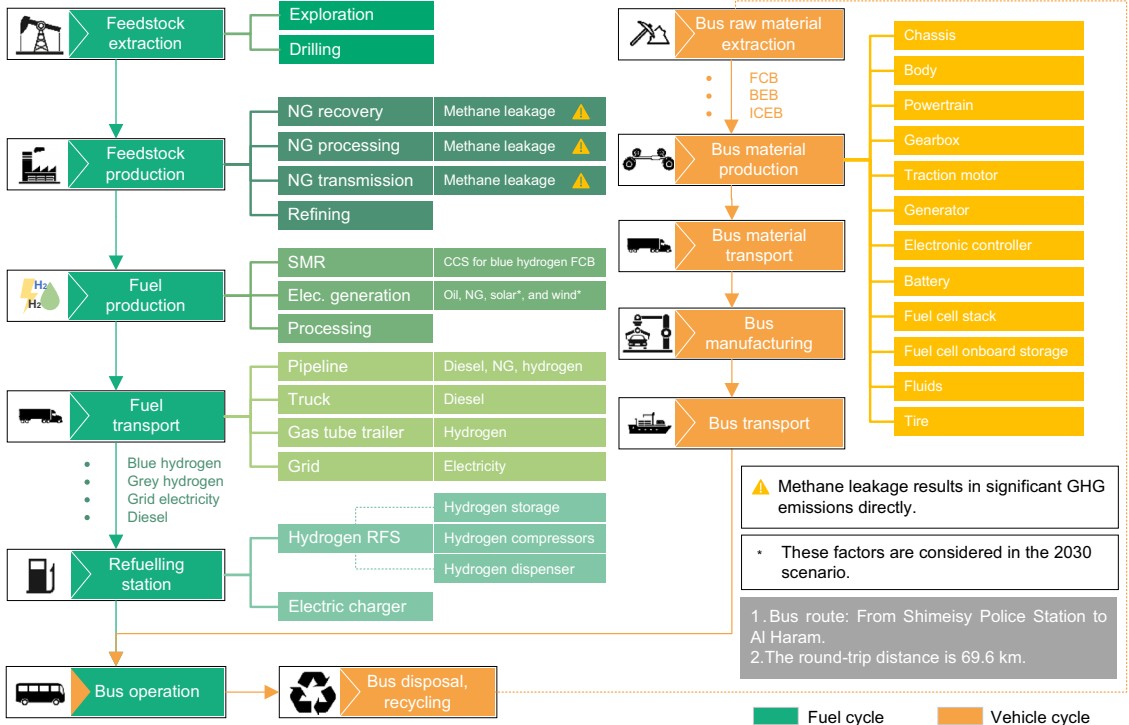

**Fig. 1 | A scope of a bus system's entire life-cycle, encompassing the fuel cycle and the vehicle cycle.** The fuel cycle is represented by green, encompasses processes such as natural gas (NG) extraction, steam methane reforming (SMR), carbon capture and storage (CCS), and distribution via refuelling stations (RFS). The vehicle cycle is represented by orange, includes different bus types like fuel cell bus (FCB), battery electric bus (BEB), and internal combustion engine bus (ICEB). The intersection of these two cycles occurs in the bus operation phase, with its greenhouse gas (GHG) emissions and energy use accounted for the fuel cycle in this study. Elec. generation stands for the electricity generation.

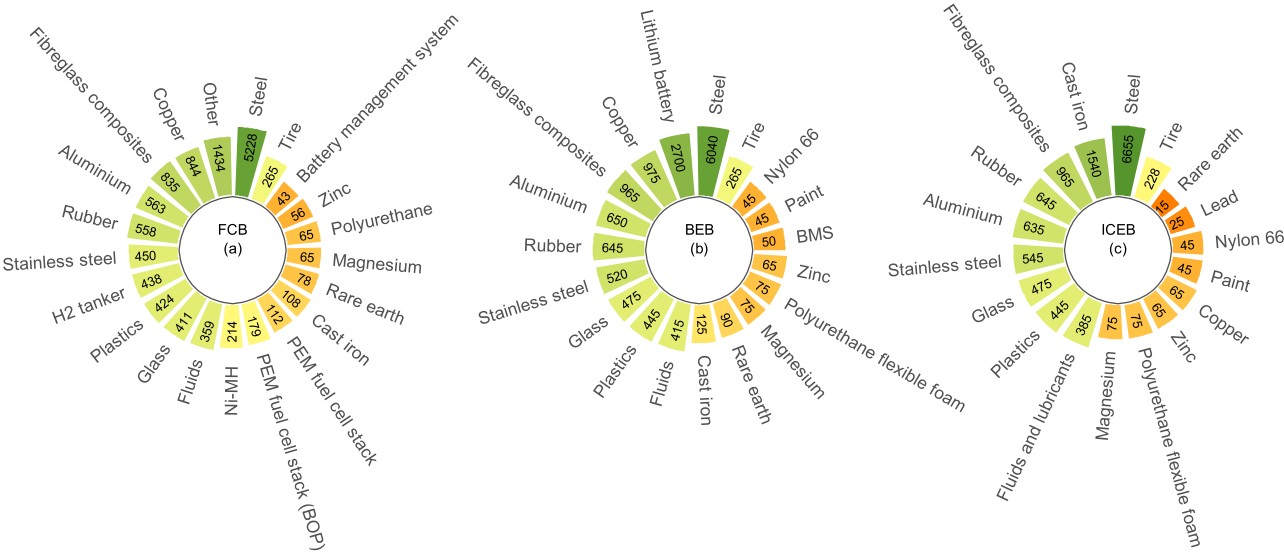

**Fig. 2 | The material composition (in kg) of proton-exchange membrane (PEM) fuel cell bus (FCB), battery electric bus (BEB), and internal combustion engine bus (ICEB). a–c** represent PEM FCB, BEB, and ICEB, respectively. Ni-MH, BOP, and BMS stand for nickel-metal hydride battery, balance-of-plant, and battery management system, respectively.

**Fig. 3 | Greenhouse gas (GHG) emissions and energy footprint (2022).** An inclusive quantification of both energy consumption and GHG emissions for three distinct bus categories. **a, b** represent the energy consumption and GHG emissions of the vehicle cycle, respectively. **c, d** illustrate the energy consumption and GHG emissions, of the fuel cycle, respectively. Proton-exchange membrane (PEM) fuel cell buses (FCBs) utilise blue or grey hydrogen, battery electric buses (BEBs) rely on grid electricity, and internal combustion engine buses (ICEBs) employ diesel fuel. * ADR assembly, disposal, and recycling, RFS refuelling station.

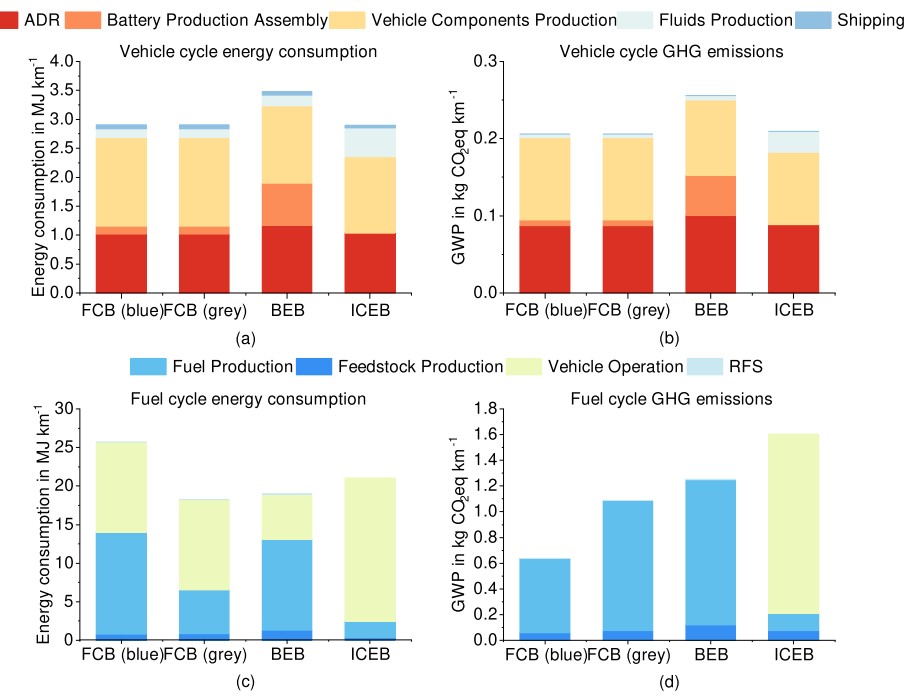

## Vehicle cycle results in 2022 scenario

The bill of materials is presented in Fig. 2. The provided material breakdown of FCBs addresses the existing gap in the research.

As depicted in Fig. 3, the comparative analysis of GHG emissions and energy consumption is presented, encompassing both the vehicle cycle and fuel cycle. In the vehicle cycle, BEBs exhibit the highest energy consumption and GHG emissions (Fig. 3a, b). This is mainly attributed to the energy-intensive nature of lithium iron phosphate (LFP) battery manufacturing, which accounts for ~21% of energy consumption and 20% of GHG emissions. A critical aspect to consider is the inefficiency in energy conversion during LFP battery production, which currently stands at a mere 0.31% (equivalent to 1176.57 MJ kWh$^{-1}$$_{LFP\ battery}$). In addition, BEBs have the highest vehicle weight, which is proportional to energy consumption and emission in the vehicle assembly, disposal, and recycling (ADR) phase.

ICEBs achieved the lowest energy consumption levels at 2.90 MJ km$^{-1}$ and relatively low emissions at 0.21 kgCO$_2$-eq km$^{-1}$, but FCBs' results are very close. The slight elevation in FCBs' energy consumption stems from the stages of nickel-metal hydride (NiMH) battery production, vehicle shipping, and vehicle components production, which are 3873%, 31%, and 16% higher than those of ICEBs, respectively, despite ICEBs' much higher value in fluid production stage. FCBs utilise NiMH batteries with a 235 kWh capacity, characterised by a comparatively lower primary energy consumption for cell production and assembly, recorded at 306.46 MJ kWh$^{-1}$$_{NiMH\ battery}$.

**Fig. 4 | Energy consumption, greenhouse gas (GHG) emissions from 'tank-to-wheel', and cooling load of air conditioning (AC) system of proton-exchange membrane (PEM) fuel cell bus (FCB), battery electric bus (BEB), and internal combustion engine bus (ICEB). a** The aggregated energy consumption and corresponding GHG emissions throughout bus operation. **b** A quantification of the cooling load generated by the various heat sources of the bus AC system, denoted in kW.

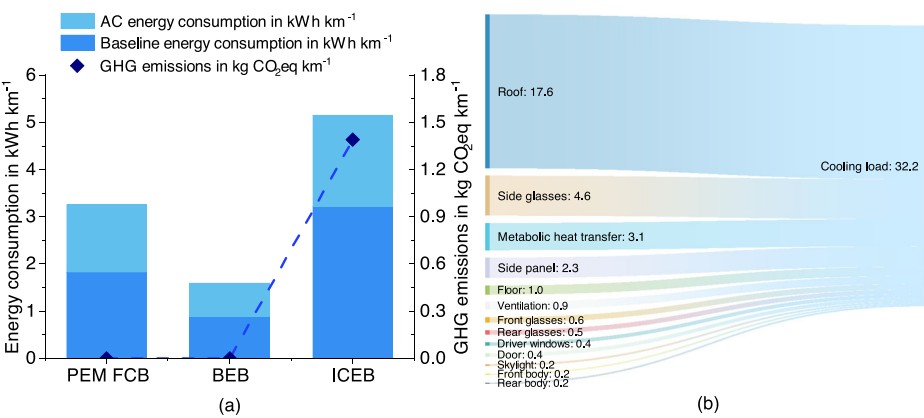

**Fig. 5 |** Analysing the well-to-wheel (WtW) greenhouse gas (GHG) emissions and energy use of blue and grey hydrogen production across fuel production, feedstock production, and vehicle operation stages through a multi-scenario evaluation of steam co-production, and its utilities in carbon capture and storage (CCS) and chemical facilities.

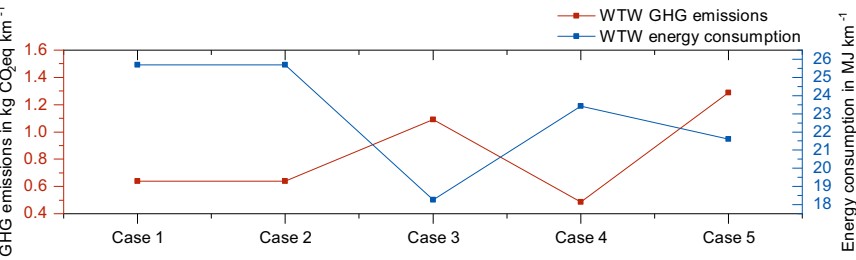

The augmented energy requirement for FCB transportation is specific to this study's assumptions. FCBs are presumed to be shipped from Japan, entailing a longer journey compared to ICEBs, which are transported from Sweden. Transportation of these buses was facilitated using a 6500 dwt cargo roll-on/roll-off vessel, optimised for a transit range of 8036 nautical miles, the maximum transport distance considered in this study. Consequently, shipping distances for hydrogen FCBs, BEBs, and ICEBs are estimated at 8036 nautical miles, 6500 nautical miles, and 6009 nautical miles, respectively.

## Fuel cycle results in 2022 scenario

The corresponding fuels for FCBs, BEBs, and ICEBs are blue and grey hydrogen, electricity from the grid, as well as diesel, respectively. The fuel cycle includes four components, with fuel production and vehicle operation dominating the contribution (Fig. 3c, d).

GHG emissions associated with the fuel cycle (hydrogen and electricity) are influenced by methane leakage during NG production. For this study, a cumulative methane leakage rate of 0.38% was assumed, a figure that stands relatively low in comparison to other regions, thereby contributing to reduced GHG emissions.

Blue hydrogen exhibits the highest energy consumption, surpassing grey hydrogen by 41%, which is the lowest-ranked. The primary distinction between the hydrogen fuels arises from CCS which reduces GHG emissions at the cost of consuming energy in the form of NG. Grey hydrogen production can generate excess steam (213.34 kJ MJ$^{-1}$ hydrogen), assuming to be exported for use in nearby chemical plants (the same amount of energy is used for CCS in blue hydrogen production)[33]. For the 2022 scenario, a CCS efficiency of 90% was assumed in this study, consistent with European technology levels[32]. The H2A model was used to estimate the energy required for $CO_2$ capture, which is 355 kWh per MJ of carbon. The calculations regarding the efficiency of $CO_2$ capture can be found in Supplementary Note 3.

A potential strategy for reducing life-cycle emissions could be transitioning from existing blue hydrogen facilities in Jubail to grey hydrogen plants in Yanbu. The energy consumption of electricity and diesel sit in the middle between the two hydrogen fuels. Bus operation dominates diesel's total consumption while electricity has the majority of its energy consumed in the fuel production phase.

On the other hand, blue hydrogen ranks the lowest in fuel cycle emissions, measuring 0.64 kgCO$_2$-eq km$^{-1}$, followed by grey hydrogen (1.09 kgCO$_2$-eq km$^{-1}$) and electricity (1.25 kgCO$_2$-eq km$^{-1}$) (Fig. 3c). The primary contributors to GHG emissions for hydrogen (blue: 91%; grey: 93%) and electricity (90%) are attributed to fuel production, as no GHG emissions are generated during the bus operation phase. In contrast, diesel demonstrates the highest emissions at 1.6 kgCO$_2$-eq km$^{-1}$, primarily due to bus operation accounting for 87% of the emissions, distinguishing it from the other fuels.

Feedstock production refers to the extraction and processing stage of NG and oil. Its contribution is minimal for all fuels in this study. The highest percentages are observed for BEBs, a result of 0.12 kgCO$_2$-eq km$^{-1}$. RFS construction's share in all fuel cycles is even smaller. Given the widespread popularity of diesel RFS, this study only examined the construction of infrastructure for new hydrogen RFS and super-fast charging stations (SFCS). Furthermore, the construction of SFCS infrastructure consumes more energy and generates higher $CO_2$-eq emissions compared to those for the hydrogen RFS infrastructure (Supplementary Notes 3 and 4).

In bus operation alone (Fig. 4a), BEBs show the lowest energy consumption of 1.60 kWh km$^{-1}$, followed by FCBs. ICEBs consume much more energy due to their lower energy conversion efficiency associated with the powertrain (30%) (for detailed information on energy conversion efficiency for all buses, refer to Supplementary Note 2). It is observed that 38–44% of the energy is consumed by the AC system during bus operation. In most LCA models and literature, the energy consumed by heating or cooling has not been considered[11,31]. However, considering the extremely hot weather conditions in Makkah, with an average annual temperature of 39 °C, the AC system needs to operate continuously while the bus is in operation. Although the energy consumption by AC did not change the overall hierarchy, the substantial amount needs to be considered for a comprehensive LCA.

Figure 4b illustrates the distribution of the AC cooling load within the passenger cabin, with the roof contributing the highest percentage of 55% to the overall thermal load. Given identical bus surface areas and consistent

Fig. 6 | Complete life-cycle results for bus energy expenditure and greenhouse gas (GHG) emissions in 2022 and 2030: incorporating all inclusive phases of both vehicle and fuel cycles, alongside outcomes for battery electric buses (BEBs) powered entirely by renewable energy. **a** Energy consumption (MJ km$^{-1}$) in 2022, **b** GHG emissions (kg km$^{-1}$) in 2022, **c** Energy consumption (MJ km$-1$) in 2030, and **d** GHG emissions (kg km$-1$) in 2030. RFS refuelling station, ADR assembly, disposal, and recycling, FCB fuel cell bus, ICEB internal combustion engine bus.

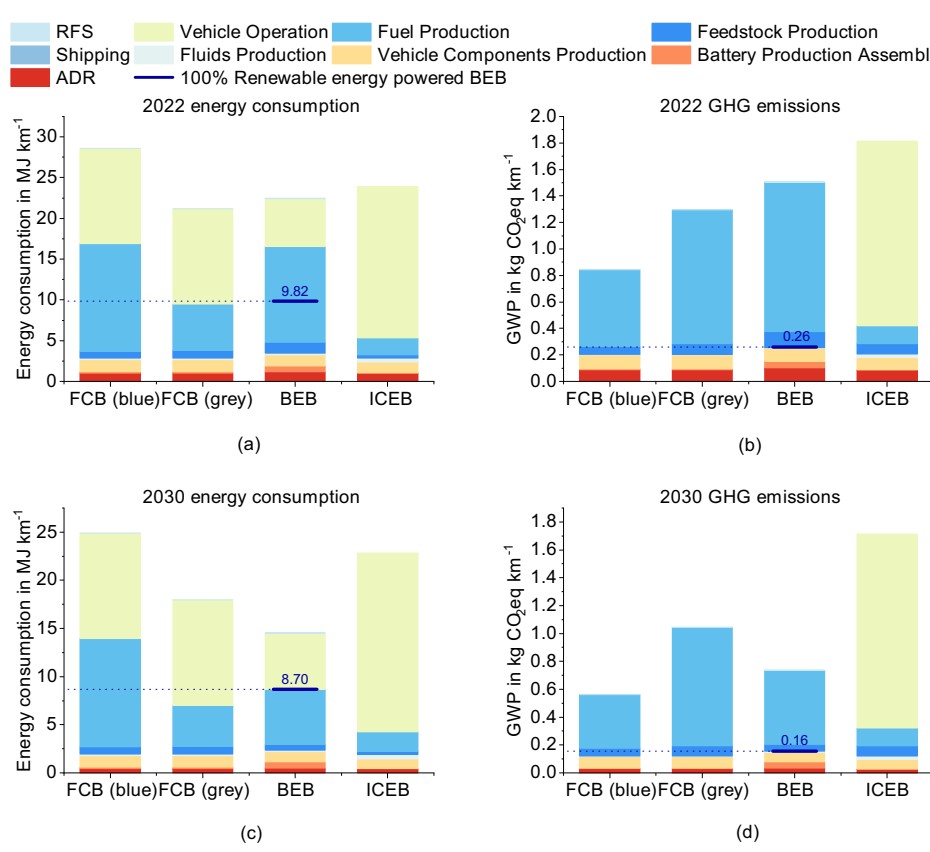

**Hydrogen production scenarios**

Case 1 is the default case for blue hydrogen production with concurrent steam co-production in Jubail. Case 2 is stand-alone blue hydrogen production in Jubail without exporting steam. Case 3 is grey hydrogen production in Yanbu using surplus steam (default case). Case 4 is blue hydrogen production in Yanbu with the recycling of excess steam. Case 5 is independent grey hydrogen production in Yanbu without repurposing surplus steam.

As delineated in Fig. 5, a comparison of GHG emissions and energy consumption across the five outlined scenarios is provided. A notable observation is the identical outcomes for Case 1 and Case 2, underscoring the hypothesis that surplus steam serves to feed the CCS, with no excess steam available for export.

A juxtaposition between cases 3 and 5 lays bare the implications of steam co-production. A fraction of the steam, carrying an energy quotient of 0.21 MJ per MJ of hydrogen produced, is dispatched to the auxiliary chemical facility. This process subsequently results in the dismissal of the corresponding energy consumption and resultant emissions from the scope of this analysis. Consequently, the GHG emissions and energy consumption recorded for case 3 exhibit a reduction of 0.20 kg km$^{-1}$ and 3.36 MJ km$^{-1}$ respectively, in comparison to case 5. The influence of steam co-production is thus observed to be considered in the context of grey hydrogen production.

Comparative analysis between cases 1 and 3 reveals that case 1, marked by the blue hydrogen production at Jubail, leads to a decrement in CO$_2$-eq emissions by 0.45 kg km$^{-1}$ in relation to case 3 (production of grey hydrogen at Yanbu). However, this outcome is accompanied by an adverse effect: the energy consumption for case 1 escalates by 7.4 MJ km$^{-1}$, in comparison to case 3. This can be attributed to the energy being wholly consumed by the CCS system.

The well-to-wheel outcomes in hydrogen production encompass a transportation phase. With an assumption that blue hydrogen can also be produced at Yanbu, thereby ensuring identical hydrogen transportation distances, a comparison of blue and grey hydrogen production yields interesting insights. By analysing cases 3 and 4, it is discerned that the blue hydrogen production in Yanbu results in a reduction of GHG emissions by 0.61 kg km$^{-1}$ and energy consumption by 5.2 MJ km$^{-1}$, relative to the grey hydrogen production.

Lastly, the findings for case 4 demonstrate much lower values compared to those of case 1. This indicates that the blue hydrogen production at Yanbu achieves a decrease in GHG emissions and energy consumption by 0.15 kg km$^{-1}$ and 2.27 MJ km$^{-1}$, respectively, in transportation, compared to the blue hydrogen production at Jubail.

**Comparison of 2022 and 2030 scenario**

Currently, the Makkah region faces a considerable deficit in renewable energy infrastructure, presenting challenges for the deployment of BEBs that depend solely on renewable energy. This study investigates not only a feasible bus fleet model but also explores a speculative scenario in which BEBs are powered entirely by renewable energy sources (100% green BEBs) for the years 2022 and 2030, aiming to ascertain their ultimate carbon mitigation efficiency relative to FCBs utilising blue hydrogen. Figure 6 provides a comprehensive portrayal of life-cycle energy consumption and GHG emissions for anticipated bus fleets in 2022 and 2030, alongside the total life-cycle effects of a conjectural fleet of BEBs powered purely by solar energy, discounting contributions from photovoltaic infrastructure.

The integrated life-cycle energy consumption and CO$_2$-eq emissions for the 2022 scenario are represented in Fig. 6a, b. Figure 6a illustrates the aggregated energy consumption per kilometre (MJ km$^{-1}$) for bus driving in 2022, while Fig. 6b graphically delineates the cumulative CO$_2$-eq emissions per kilometre (kg km$^{-1}$) for the same period. In terms of the entire bus life cycle, the vehicle cycle has a relatively minor influence on both energy consumption (10–15%) and CO$_2$-eq emissions (12–24%) compared to the

**Table 1 | Parameters of the 2030 and 2022 scenarios comparison**

| Parameters | | 2022 | 2030 |
|---|---|---|---|
| 1. Energy mix | Oil | 39.2%[57–59] | 0.0%[4,34] |
| | NG | 60.6%[57–59] | 50.0%[4,34] |
| | Solar | 0.2%[57–59] | 36.4%[4,34] |
| | Wind | 0.0%[57–59] | 13.6%[4,34] |
| 2. CCS efficiency | | 90.0%[32] | 96.2%[55] |
| 3. Shale gas share | | 0% | 9.7%[48,49] |
| 4. Bus weight in kg | FCB | 12,464[12,125–127] | 9939[128] |
| | BEB | 14,400[12,125–127] | 11,483[128] |
| | ICEB | 12,700[12,125–127] | 9457[128] |
| 5. Recycled steel[b] | FCB | 23.2%[35] | 73.6%[33] |
| | BEB | 10.1%[35] | 73.6%[33] |
| | ICEB | 41.6%[35] | 73.6%[33] |
| 6. Recycled aluminium[b] | FCB | 99.4%[36] | 64.9%[33] |
| | BEB | 19.2%[36] | 64.9%[33] |
| | ICEB | 70.9%[36] | 64.9%[33] |
| 7. Recycling rate of $Li_2CO_3$ | | 0% | 30.0%[c] |
| 8. FC stack efficiency | | 52.0%[37,129] | 61.0%[37] |

[a]The weight of the bus does not include the weight of the tyres.
[b]The percentage of recycled materials utilised in the production of buses.
[c]This is predicated on our hypothesis.

fuel cycle. The observed ranking trends for energy consumption and $CO_2$-eq emissions align with the findings derived from the fuel cycle analysis.

In the 2022 scenario, blue FCBs exhibit the lowest life-cycle emissions at 0.84 $kgCO_2$-eq $km^{-1}$, followed by grey FCBs (1.29 $kgCO_2$-eq $km^{-1}$), BEBs (1.51 $kgCO_2$-eq $km^{-1}$), and ICEBs (1.81 $kgCO_2$-eq $km^{-1}$). Comparing blue FCBs to ICEBs, carbon emissions were reduced by 53.6%, while energy consumption increased by 19.5%. Conversely, BEBs exhibited a lower emission reduction of only 16.9%, but also achieved a decrease in energy consumption by 6.1%. The majority of ICEBs emissions originate from bus operation, whereas fuel production dominates the other three life cycles. The hierarchy of energy consumption differs from the emission results, with minimal variations among all options. Blue FCBs rank highest at 28.59 $MJ km^{-1}$, primarily due to the energy-intensive CCS process. ICEBs rank second, with substantial energy consumption occurring during bus operation. Despite having the highest energy consumption within their vehicle cycle, BEBs demonstrate the second lowest overall consumption. This is due to their high fuel cycle energy conversion efficiency. Interestingly, grey FCBs exhibit the lowest life-cycle energy consumption. In a hypothetical scenario where BEBs are powered entirely by solar energy, the forecasted energy consumption and $CO_2$-eq emissions are expected to decrease to 9.82 $MJ km^{-1}$ and 0.26 $kgCO_2$-eq $km^{-1}$, correspondingly. This denotes a 54% reduction in energy usage compared to grey hydrogen FCBs and a 69% decrease in GHG emissions relative to blue hydrogen FCBs.

The 2022 scenario represents the current conditions in Saudi Arabia, while a 2030 scenario is created to examine potential changes in the LCA results in the near future. Table 1 presents the modified parameters for 2030, along with their corresponding values used in 2022.

In the Saudi Green Initiative's 2030 vision, the share of renewable energy would increase to 50% by 2030[4]. Based on energy administration data, the projected energy mix for 2030 envisions NG, solar, and wind accounting for 50%, 34.4%, and 13.6% of the total energy sources, respectively, as depicted in Supplementary Note 4[34].

In the blue hydrogen cycle, the current CCS efficiency stands at 90% in 2022, and it is hypothesised to further improve to 96.2% by 2030 (Supplementary Note 3). Regarding NG production, it is estimated that 9.7% of the NG supply in 2030 will come from shale gas. Furthermore, as vehicle light-

weighting is expected to be a trend in the future, it was assumed that by 2030, 30% of the steel in vehicles will be replaced by aluminium. The estimated weights of buses in 2030 are listed in Supplementary Note 6.

The assembly of vehicles in 2022 primarily takes place overseas, thereby necessitating the utilisation of global data for metal material recycling in assembly processes. The estimated shares of recycled steel employed in vehicle manufacturing in Japan, China, and Sweden stand at 23.2%, 10.1%, and 41.6%, respectively[35]. Furthermore, the usage of recycled aluminium in these respective countries is noted to be 99.4%, 19.2%, and 70.9%[36]. Regarding LFP batteries recycling, since there are currently no battery recycling facilities in the Middle East, a recycling rate of 30% for $Li_2CO_3$ was assumed using the hydro-metallurgical process with organic acid leaching in 2030, as employed in GREET.

In forecasting the scenario for the year 2030, this study incorporates data from GREET to forecast the use of recycled materials in the assembly process. Specifically, it is projected that the share of recycled steel used in vehicle manufacturing will be 73.6%, while the share of recycled aluminium will reach 64.9%.

In the 2030 scenario, it was assumed that all vehicles will be assembled within Saudi Arabia, eliminating the need for shipping and reducing energy consumption and emissions associated with transportation. Considering that ICEBs and BEBs already operate at highly efficient levels, it is hypothesised that the FC stack efficiency for FCBs will increase from 52% to 61% by 2030[37].

Figure 6c presents a detailed quantification of total energy consumption per kilometre ($MJ km^{-1}$) for bus operation in 2030. Concurrently, Fig. 6d provides an estimation of the total life-cycle $CO_2$-eq emissions per kilometre ($kg km^{-1}$) for bus driving projected for the same year. The results of the 2030 scenario (Fig. 6c, d) suggest a widespread decline over 2022, however, the extent of decline of individual option varies, which leads to hierarchy changes. BEBs show the largest decline, with emissions decreasing by 51% and energy consumption decreasing by 35%, making them the least energy-consuming and the second least emitting option. The substantial majority of the contraction in emissions (87%) and energy consumption (86%) derive from the fuel cycle, owing to the notable increase in renewable energy shares in Saudi Arabia's power supply.

Blue FCBs still offer the lowest life-cycle emissions in the 2030 scenario, but their leading edge over BEBs shrinks largely. The reduction in emissions primarily derives from the fuel cycle, mainly attributable to the boosted CCS efficiency and the increment in the share of renewables for power supply. Grey FCBs exhibit a moderate decline of 19% in emissions and 15% in energy consumption, losing their position as the least energy-consuming option to BEBs. The primary emission reductions can be attributed to changes in the fuel cycle resulting from a shift in the energy mix towards renewables. However, the impact of this change is relatively less pronounced for grey FCBs compared to BEBs.

ICEBs remain the most emitting option among all bus categories, demonstrating a marginal decline. Unlike FCBs and BEBs, the principal reduction appears within the vehicle cycle, attributed to the implementation of vehicle light-weighting and the increased usage of recycled materials.

Remarkably, a hypothetical 100% green BEBs scenario emerges as the most GHG efficient model in 2030, achieving GHG emissions that are 79% lower than grid-connected BEBs and 72% lower than purely blue hydrogen FCBs and surpassing the grid-connected BEB by 40% in energy efficiency.

For 2030, FCBs powered by a blend of 98% blue and 2% grey hydrogen are projected to produce $CO_2$-eq emissions of 0.57 $kg km^{-1}$ and require 24.74 $MJ km^{-1}$ of energy, paralleling the environmental impact of FCBs powered solely by blue hydrogen.

Supplementary Information provides detailed life-cycle results for various bus models for the 2022 and 2030 scenarios, both in base and functional units, as presented in Tables 14, 16, and 17.

**Sensitivity analysis of GHG emissions**

Figure 7 delineates the differential impacts elicited by parametric adjustments on the GWP100 for the year 2022, which elucidates the dichotomy

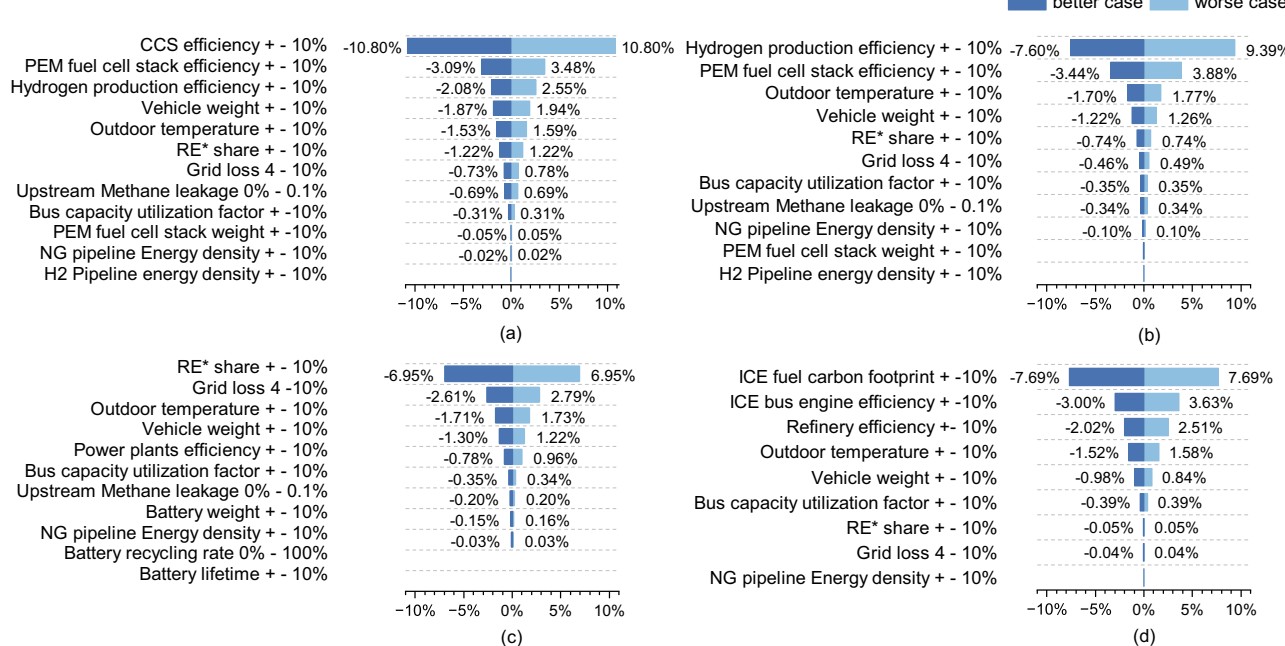

**Fig. 7 | Comparative analysis of the 100-year global warming potential (GWP100) in 2022.** This analysis contrasts a 'worst-case' amplification and 'better-case' mitigation in blue (**a**) and grey (**b**) proton-exchange membrane (PEM) fuel cell buses (FCBs), battery electric buses (BEBs) (**c**), and internal combustion engine buses (ICEBs) (**d**). **a**–**d** illustrate the impact of a ±10% factor on GWP100, with exceptions for certain ranges. *Renewable energy (RE) adjustments are based on a 50% renewable energy baseline for the energy mix in the grid. CCS carbon capture and storage, NG natural gas, ICE internal combustion engine.

between the exacerbation observed in a 'worst-case' scenario and the mitigation realised in a 'better-case' scenario, consequent to alterations in various parameters. Specifically, Fig. 7a–d expound upon the repercussions of these adjustments on distinct vehicle categories: blue FCBs, grey FCBs, BEBs, and ICEBs, respectively.

Blue FCBs' life-cycle emissions are most sensitive to CCS efficiency, with a 10% variation inducing 10.8% change (Fig. 7a). Factors such as PEM stack efficiency, hydrogen production efficiency, vehicle weight, outdoor temperature, and grid's renewable energy share exert milder impacts, affecting emissions by 1–4%. For grey FCBs, hydrogen production efficiency is the most sensitive factor (Fig. 7b), followed by PEM stack efficiency and outdoor temperature. Renewable energy's share influences BEBs the most, with a 10% increase causing a 6.95% decrease in emissions (Fig. 7c). Additional factors have mild impacts, encompassing grid losses, outdoor temperature, and vehicle weight. Lithium battery-related factors show less-than-expected impacts, because replacement is not required within the life-cycle (Supplementary Note 1) For ICEBs, diesel's carbon footprint predominantly dictates the overall impact. This finding underscores the importance of reducing fuels' carbon footprint and highlights the need for further exploration of low-carbon fuel solutions. Additional factors, such as engine efficiency, refinery efficiency, and outdoor temperature influence emissions to a lesser extent.

In assessing the impact of dynamic passenger loads in transit systems, this study introduces the concept of the "bus capacity utilisation factor". Defined as the ratio of actual passenger count to maximum seating capacity (Supplementary Table 1), this factor was adjusted by ± 10% during sensitivity analysis. As illustrated in Fig. 7, the investigation reveals that variations in this factor have a marginal influence on the operational efficiency of the evaluated bus types.

### Results of other environmental impacts
The GHG emissions in $CO_2$-eq presented so far are associated with Global Warming Potential over a 100-year time frame (GWP100). We have also investigated emissions with GWP20, acidification potential (AP),

eutrophication potential (EP), and photochemical oxidation potential (POP) in the 2022 scenario (Fig. 8).

There is no notable difference between GWP20 and GWP100 results, except that emissions associated with feedstock production are observed to be slightly higher in GWP20, particularly for BEBs and ICEBs. BEBs perform the worst in AP, EP, and POP assessments, while grey FCBs rank the best. The results highlight the potential environmental challenges facing BEBs, primarily attributed to $NO_x$ emissions from fossil fuel for electricity generation, which could be improved by switching to renewable energy or other clean power. The environmental impacts for the 2030 scenario are detailed in Supplementary Note 7.

### Conclusion
When considering feasible fuel supply, our study identified blue FCBs as the leading solution for decarbonising the urban bus transportation sector within the short-to-medium term. The life-cycle GHG emissions results highlight that the replacement of ICEBs with blue hydrogen FCBs can cut emissions by almost half if deployed for public transportation immediately even considering higher energy consumption. Our results indicate promising prospects for deeper emission cuts by 2030, driven by technological advancements and the expected shift towards renewable energy sources. The performance of 2030 mixed hydrogen scenario is nearly identical to pure blue hydrogen scenarios. The sensitivity analysis emphasises the urgency for enhancements in CCS technologies, hydrogen production efficiency, and FC stack efficiency.

Contrarily, our results demonstrate that when considering a feasible electricity supply, BEBs in the current 2022 scenario do not notably contribute to GHG emissions reduction across the entire bus life-cycle. This is primarily due to the power sector's continued reliance on fossil fuels in Saudi Arabia, coupled with the LFP battery's considerable carbon footprint. The sub-optimal performance of BEBs on other crucial environmental impacts further questions BEV as a climate-friendly solution. Nevertheless, future scenarios envision substantial reductions in the life-cycle GHG emissions of BEBs by 2030. These anticipated improvements hinge on the increment of

**Fig. 8 | Life-cycle impact analysis (LCIA) results showcasing the 20-year global warming potential (GWP20), eutrophication potential (EP), acidification potential (AP), and photochemical oxidation potential (POP) calculated using the CML2001 methodology.** This comprehensive analysis spans the entire life cycle of buses in 2022, comparing blue and grey hydrogen fuel cell buses (FCB), battery electric buses (BEB), and internal combustion engine buses (ICEB). **a–d** illustrate the impacts on GWP20, EP, AP, and POP, respectively. RFS stands for refuelling station, HVAC stands for heating, ventilation, and air conditioning.

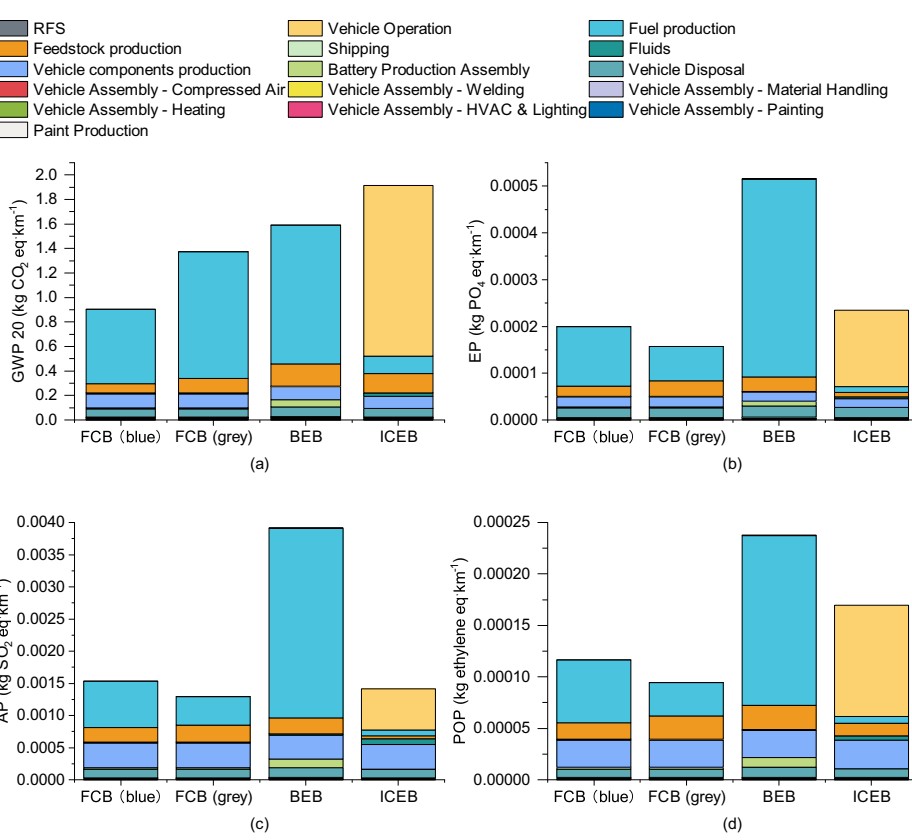

renewable energy's share and advancements in vehicle light-weighting, making BEBs a competitive decarbonisation solution to FCBs in 2030.

In a theoretical framework, BEBs powered entirely by renewable energy emerge as the premier option for carbon reduction and energy efficiency. Nevertheless, the practicality of exclusively using renewable energy for BEBs depends on factors such as the availability of renewable resources and Saudi Arabia's strategic progress in the renewable energy sector, especially the allocation of renewable energy to the transport sector. While BEBs powered by renewable sources exhibit zero emissions in the fuel cycle, the wider environmental and energy impacts associated with the infrastructure for renewable energy generation, including photovoltaic systems, necessitate thorough investigation.

Under all circumstances, ICEBs bear the highest life-cycle carbon footprints, a consequence of the immense emissions generated during bus operation. Therefore, it becomes increasingly clear that curbing GHG emissions necessitates an essential focus on reducing the carbon footprint of fuels, highlighting the need for low-carbon fuels, like "e-fuels". Nevertheless, the nascent stage of these fuels necessitates further LCA studies to deepen our understanding of the decarbonisation potential inherent in the overall system. Furthermore, an evaluation of the energy efficiency of these alternatives is indispensable, ensuring a holistic approach to their integration into the energy landscape.

A comprehensive environmental assessment and energy consumption of the life-cycle of urban buses was conducted in this study, specifically including aspects such as the bus transportation phase (shipping), RFS infrastructure construction, and the performance of bus AC systems operating under high-temperature conditions. This study employed a combination of open-source models, original models and data, various databases, and LCA methodologies. Although it is a tailored case study, the insights of this work offers a potential decarbonisation pathway pertinent to non-OECD countries, with the developed model serving as a universally applicable tool for any country to evaluate and tailor its unique decarbonisation strategy.

In the context of LCA, our results emphasise the value of a wider scope when evaluating the environmental impact. Such evaluations yield more comprehensive views and deeper insights of the emission reduction potential and sustainability performance of new technologies, products, or processes. Furthermore, achieving sustainable bus transport solutions necessitates a multifaceted strategy, extending beyond environmental impacts to include economic assessments of bus alternatives and examination of boundary conditions like the efficacy of CCS in the blue hydrogen cycle, carbon sequestration capabilities, and the development of renewable energy infrastructure.

## Methodology
### Functional unit
In the framework of LCA, the "functional unit" is a crucial metric that quantifies the functionality of the products under investigation. Specifically, in evaluating bus systems, they are classified into two cycles: fuel cycle and vehicle cycle. To methodically assess energy consumption and emissions, the functional unit is defined as "1 kilometre travelled by a bus" under general operational conditions. Public transport systems exhibit passenger number fluctuations due to temporal and seasonal factors. To mitigate the impact of such variations, this investigation employs a hypothetical scenario where three bus models, each with a capacity for 30 passengers, operate under uniform load conditions. This approach ensures an equitable comparison of life-cycle emissions across the bus models, effectively neutralising the discrepancies arising from dynamic passenger numbers.

### Refined methodology for energy and emissions calculation
This study employed a bottom-up LCA methodology, adhering to ISO 14040–14044 (2006) standards, to determine life-cycle emissions and energy usage across various bus technologies and their corresponding fuel systems[38,39]. The core calculations were conducted by utilising the GREET 2022 model from the National Renewable Energy Laboratory[33]. A critical point to consider, however, is the absence of bus-related parameters within

the confines of GREET model. To counter this, a series of assumptions were made based on the company consultancy, underpinned by a comparison of structural features between passenger cars and buses. For instance, the GREET model linked passenger vehicles' energy consumption and emissions with vehicle weight during the energy and emissions evaluation of FCVs, BEVs, and ICEVs in the ADR phase. Extending this, it was assumed that the energy use and emissions of buses in the ADR phase align proportionately with passenger cars. Consequently, the energy and emissions footprints of FCBs, BEBs, and ICEBs in the ADR phase were considered to be proportional to their weight.

Original models, such as for hydrogen RFS (Supplementary Note 3), SFCS, roll-on/roll-off vessel transport, and bus AC energy consumption and emissions were developed using life-cycle coefficients procured from GREET. The hydrogen analysis (H2A) production model facilitated the simulation of grey and blue hydrogen production. Furthermore, a Python-based model was employed to estimate average inter-site distances at the country level between power and fuel production plants for the electricity cycle, with geographic data sourced from open-access geographic information system databases (see Supplementary Note 4). These primary data were integrated into the GREET model for the entire life-cycle calculations.

### Life-cycle inventory data collection and development

Life-cycle inventory data combined primary sources, secondary sources, and assumptions, supplemented by company consultations, literature reviews, and databases such as GREET and ecoinvent version 3.9.1[33,40]. A detailed description of the inventory data is provided in Supplementary Information.

The material composition excluding tyres of BEBs and ICEBs was sourced from the extant literature[12]. The composition of FCB was determined by integrating information from existing literature, company consultation, and other publicly accessible data sources.

### LCIA methodology

A life-cycle impact analysis (LCIA) was conducted to evaluate all the environmental impacts, including the GWP100, GWP20, AP, EP, and POP, employing the CML2001-LCIA methodology. Supplementary Note 7 incorporates the related LCIA characterisation factors (Supplementary Table 13) as well as the results of these environmental impacts for the year 2030 (Supplementary Fig. 6).

### Sensitivity analysis

In this study, a sensitivity analysis was conducted to examine the varying effects of distinct impact factors on GWP100 for different buses. We systematically adjusted nearly all impact factors within the range of −10% to +10% from the default case. However, grid losses and upstream methane leakage rate exhibited distinct considerations. Grid losses were subjected to adjustments within the 4−10% range, while maintaining the default value of 7%, based on company consultations. Regarding upstream methane leakage rates, we explored a range spanning from 0% to 1%, while retaining a default value of 0.05%[41]. To establish a standardised framework, the scheme resulting in an increased GWP100 was defined as the "worse case", while the scheme leading to a decreased GWP100 was termed the "better case".

### Hydrogen cycle

**Hydrogen cycle scope.** In the 2022 scenario, based on data from hydrogen production facilities within Saudi Arabia, this study hypothesised that blue hydrogen was derived from Jubail, while grey hydrogen originated from Yanbu[42,43]. Both blue and grey hydrogen were transported from production facilities to bulk terminals via pipelines, then transported to Makkah's hydrogenation stations using tube trailers. In both cases, blue and grey hydrogen were produced from NG by the SMR process. The energy density data for hydrogen pipelines and tube trailers is derived from the GREET model, which is detailed in Supplementary Note 9.

The primary distinction between the production of blue and grey hydrogen lies in the utilisation of CCS technologies (with the CCS efficiency of 90%[32]), the locations of NG fields, and the positions of hydrogen production factories in blue hydrogen production. Blue hydrogen relied on NG from the Fadhili gas field, transported via a pipeline spanning 74.9 km, with an energy intensity of 567 BTU ton$^{-1}$mile$^{-1}$ [44]. In contrast, grey hydrogen utilised NG transported through Saudi Arabia's longest pipeline, extending from Abqaiq in the east to Yanbu in the west, a total distance of 1193 km.

In this study, hydrogen production is solely dependent on conventional NG sources, as unconventional gas extraction methods, such as shale gas, remain largely untapped in Saudi Arabia despite the country's vast reserves. The 2022 scenario envisions a complete reliance on conventional NG. Drawing from existing literature, the processing and recovery efficiency for conventional NG was assumed to be 97.4% and 97.5%, respectively[45–47]. Given the GREET model's methane content of 0.9216 $g_{CH4}$ $g^{-1}_{NG}$ in NG, the loss factor for conventional gas is 1.001.

The development of the Jafurah unconventional gas field could substantially transform this landscape. As Saudi Arabia's largest unconventional gas field, Jafurah is slated to commence production in 2025, boasting an estimated capacity of 2 billion standard cubic feet per day (scfd), which could substantially impact hydrogen production[48]. Furthermore, conventional gas production has experienced a 10% increase over the past decade[48]. Assuming this growth trend continues, total NG production capacity is projected to reach 20.72 billion scfd by 2030[48,49]. In light of these developments, the 2030 scenario envisions a blend of 9.7% shale gas and 90.3% conventional NG for hydrogen production.

Methane emissions, characterised by a high $CO_2$-equivalent conversion index, contribute substantially to GHG emissions and affect the overall GHG emissions of the hydrogen cycle. These leakages arise during various stages of NG production, processing, transmission, and distribution, and may be categorised as fugitive (unintentional), venting (intentional), or incomplete flaring (incomplete combustion)[50]. As per Aramco's 2022 report, Saudi Arabia's methane leakage rate during NG upstream production is quantified at 0.05%[41], comparatively lower than other regions. The leakage rate during NG processing in Saudi Arabia, estimated at 0.03%, is based on GREET hybrid data, while transmission and compression rates are reported at 0.18% per 1000 km by the ecoinvent database[51–54]. The Environmental Protection Agency's Greenhouse Gas Inventory suggests that approximately 0.09% of methane emissions are attributed to the NG distribution phase[53].

Supplementary Note 3 provides data on hydrogen compression efficiency, while Supplementary Table 7 identifies loss factors associated with hydrogen transport. Note that this study excluded the infrastructure construction of the SMR plant from its scope.

**Hydrogen production simulation.** The efficiency of hydrogen production varies between blue and grey methods, with the use of CCS during blue hydrogen production leading to a reduction in overall efficiency. In this study, a hydrogen SMR production plant was investigated, with a designed capacity to produce 201,000 kg of hydrogen per day. The low heating value efficiency of blue hydrogen production was found to be 69%, whereas grey hydrogen production achieved a higher low heating value efficiency of 76%, based on a report by KAPSARC[32]. Both hydrogen production methods assumed specific water consumption of 10 litres per kilogram of hydrogen and specific electricity consumption of 0.5 kWh per kilogram of hydrogen[32]. To evaluate the efficiency of various hydrogen production pathways, the H2A model was used in this study to quantify both the hydrogen production efficiency and the amount of input feedstock required. The H2A model utilised data on the daily hydrogen production volume of the production plant, as well as the electricity and water consumption per kilogram of hydrogen, and the low heating value production efficiency of hydrogen. The model also estimated the amount of NG feedstock required for hydrogen production, which is essential for assessing the environmental impact of hydrogen

production. Our simulations using the H2A model indicated that NG consumption for blue and grey hydrogen production was 0.1635 mm BTU kg$^{-1}$ hydrogen (98.97%) and 0.1483 mm BTU kg$^{-1}$ hydrogen (98.86%), respectively.

**Hydrogen production scenarios.** Presently, several SMR hydrogen plants are operational in Saudi Arabia[42]. In the SMR process, an endothermic reaction occurs, drawing heat from the reformer through the firing of catalyst tubes with a combination of recycled syngas and supplemental NG. The resulting super-heated steam produced during hydrogen generation within the SMR facility surpasses the internal process steam requirements. The excess steam emanating from blue and grey hydrogen production processes could either power the CCS system or find utility in alternative chemical factories, a manifestation commonly referred to as steam co-production[55]. Cases that do not consider exporting the surplus steam are viewed as stand-alone production. Furthermore, in this study, we have evaluated the plan of producing blue hydrogen in a grey hydrogen production plant located near Makkah to investigate the impact of reduced transportation distance on the fuel cycle of hydrogen. It is assumed that steam boilers employed for steam generation exhibit an energy efficiency of 85%[55].

**Hydrogen refuelling station infrastructure.** The materials list for the RFS inventory in this study was based on a bill of materials prepared by Mailänder in 2003 for a HydroStatoil model hydrogen RFS in Reykjavik[56]. The hydrogen PEM FCBs under study have a 600-litre hydrogen tank (equivalent to 23.46 kg of hydrogen) and consumes an estimated 9.7 kg per 100 km with the AC system turned on. The capacity of the designed hydrogen RFS was determined based on the total distance travelled by 18 buses per year, the frequency of hydrogen refuelling, and the fuel consumption per 100 km. The Makkah hydrogen RFS features a daily capacity of 259 kg, adeptly accommodating the 245 kg per day hydrogen refuelling requirements of the buses.

The detailed material composition of the infrastructure components is provided in Supplementary Table 8. Supplementary Table 9 outlines data regarding the lifetime of crucial elements within the RFS infrastructure, including RFS units, compressors, storage tanks, trailers, and hydrogen pipelines. The study assumed a 10-year lifetime for the entire bus fleet. All the components mentioned above are expected to have a service life of at least 10 years. Therefore, it can be inferred that the missing components of the fuelling station infrastructure need not be replaced within the specified 10-year period.

### Electricity cycle

**Energy mix structure in Saudi Arabia in 2022 and 2030.** Utilising a multitude of data sources[57–59], this study conducted an evaluation of Saudi Arabia's electricity generation landscape for 2022, predicated on the assumption that it remains consistent with the 2021 data. The energy portfolio is primarily composed of NG, producing 215.93 TWh, followed by oil at 139.86 TWh, and solar energy contributing a modest 0.83 TWh[57–59]. In accordance with the Saudi Green Initiative's 2030 vision, the target is to increase the share of renewable energy to 50% by 2030[4]. Predictions from the Ministry of Energy (MOE) indicate that by 2030, the expected distribution of NG, solar, and wind energy will constitute 50%, 34.4%, and 13.6% of the total energy sources respectively, as depicted in Supplementary Fig. 3[4,34].

The allocation of various power plants and their corresponding efficiencies strongly affect well-to-wheel GHG emissions of the electricity cycle. Employing annual $CO_2$ emissions data from the $CO_2$ footprint database for Saudi Arabian power plant, along with $CO_2$ emission factors for a range of power generation technologies detailed in the literature and ecoinvent database (as shown in Supplementary Table 10)[5,40,60], the electricity produced by each power plant can be determined. As a result, the technical composition of oil-fired power plants in Saudi Arabia consists of 39.3% combined cycle gas turbines, 34.2% single cycle gas turbines, 17.8% steam turbines, and 8.7% diesel generators. Conversely, gas-fired power plants feature a technology distribution of 23.6% combined cycle gas turbine NG plants, 46.8% single cycle gas turbine NG plants, and 29.6% steam turbine NG plants.

In terms of power plant efficiency in Saudi Arabia, based on the literature and the GREET database, the average efficiencies for oil-fired steam turbines, gas turbines, and diesel generators were assumed to be 32.6%, 26%, and 30%, with gas-fired steam turbines and gas turbines at 35% and 32.9%, respectively[61]. Furthermore, the average efficiency of gas-fired combined-cycle power plants was assumed to be 58.8%, while the efficiency of oil-fired combined-cycle power plants is estimated at 46.5%, grounded on the efficiency ratio of oil-fired gas turbines and gas-fired simple-cycle gas turbines, as determined by the efficiency of the gas-fired combined-cycle gas turbine[62].

The reliability of the electrical grid plays a crucial role in determining the environmental impact of electricity generation. This study estimated the grid losses to be 7%, and a sensitivity analysis was carried out to evaluate the influence of different levels of grid power losses on the final outcomes. The emission factors for distinct power plants are detailed in Table 2.

**Table 2 | Emission factors (g kWh$^{-1}$) for various power generation technologies and fuel types**

| Emissions | Oil-fired power plant | | | | Gas-fired power plant | | |
|---|---|---|---|---|---|---|---|
| | CC turbine | Steam turbine[a] | Diesel generator | Gas turbine | Steam turbine[a] | SC turbine[a] | CC turbine[a] |
| VOC | 0.0283 | 0.0113 | 0.6885 | 0.1099 | 0.0251 | 0.0171 | 0.0044 |
| CO | 0.1036 | 0.1588 | 2.1140 | 1.7568 | 0.2544 | 0.2544 | 0.0150 |
| NOx | 1.7343 | 3.3541 | 13.6142 | 4.6571 | 0.4669 | 0.4669 | 0.1739 |
| PM$_{10}$ | 0.1553 | 0.0814 | 0.8137 | 0.2898 | 0.0062 | 0.0062 | 0.0033 |
| PM$_{2.5}$ | 0.1353 | 0.1130 | 0.8318 | 0.2524 | 0.0062 | 0.0062 | 0.0033 |
| SO$_x$ | 0.5719 | 5.5351 | 0.4835 | 1.0669 | 0.0073 | 0.0073 | 0.0039 |
| BC | 0.0062 | 0.0083 | 0.1248 | 0.0151 | 0.0059 | 0.0012 | 0.0005 |
| OC | 0.0041 | 0.0058 | 0.3244 | 0.0101 | 0.0152 | 0.0276 | 0.0112 |
| CH$_4$ | 0.0257 | 0.0662 | 0.0704 | 0.0480 | 0.0124 | 0.0124 | 0.0066 |
| N$_2$O | 0.0054 | 0.0517 | 0.0690 | 0.0102 | 0.0124 | 0.0124 | 0.0067 |
| CO$_2$ | 480[b] | 1010 | 760[b] | 760[b] | 679 | 679 | 364 |

[a]The emissions factor data was primarily sourced from the Ecoinvent database and the GREET model, except for $CO_2$, for which our assumptions are based on the $CO_2$ emission factor of other power plants.
[b]The $CO_2$ emission factor was obtained from the literature[5].

**Transportation distances of electricity feedstocks.** Given that NG and oil are the primary feedstocks for electricity production, this study scrutinised the transportation phase of the raw materials for the electricity cycle. Specifically, the average distance between the power plant and the oil refinery and NG field was estimated. Location data for NG fields and oil refineries were gathered from KAPSARC, as depicted in Supplementary Fig. 1[32]. Geospatial information in the form of shapefiles containing geographic data was processed using Geopandas, and the Haversine formula was employed to compute the great-circle distance between two points represented by latitude and longitude coordinates[63]. The average distance was calculated by summing the distances between the two power sources and dividing by the number of paths. The average transport distance between oil refineries and power plants and between NG fields and power plants was found to be 742 km and 577 km, respectively.

**Super-fast charging stations.** The SFCS model used in this study is based on the literature and was adapted to maintain the systematic operation of the bus fleet[64]. The charging station comprises six chargers, three power supply units, and a control unit, enabling up to six buses to be charged concurrently with a maximum output power of 350 kW per charger. However, to ensure safety, the BEBs in this study have a charging power of 150 kW and require a full charging duration of 2–2.5 hours. Each power unit can serve up to two chargers. The control unit includes a communication unit that acts as a centralised system for power and load management of the SFCS. The composition of materials is detailed in Supplementary Table 11, with emission factors for material production sourced from the GREET database.

**Diesel cycle**

The diesel fuel cycle consists of four main steps from crude oil extraction to fuel dispensing at the retail pump stations as shown in Supplementary Fig. 4. The first step involves the extraction of crude oil from oil wells located predominantly in the Eastern Province of Saudi Arabia. The recovery efficiency of the crude oil is taken to be 99.1%. The extracted crude oil is transported via pipelines to the Abqaiq processing plant wherein the sulphur content of the crude oil is reduced for eventual transportation to refineries. Due to close proximity of the oil wells to the Abqaiq processing plant, the energy use and emissions during the transport of crude oil to Abqaiq and its processing within the plant are negligible and are not included in the life-cycle calculations. Based on the geographical location of Makkah on the western coast of Saudi Arabia, the city is supplied with diesel fuel from SAMREF and YASREF refineries in Yanbu. Therefore, the crude oil needs to be transported via the 1193 km long east-west pipeline from the Abqaiq processing plant to refineries in Yanbu. Based on the literature, the energy intensity of pipeline transport of sweet crude oil is taken as 260 BTU $ton^{-1}mile^{-1}$ [45]. Among several factors, the refinery efficiency is a strong function of the crude oil quality American Petroleum Institute (API) gravity and sulphur content) and is calculated based on the crude oil input to SAMREF and YASREF. SAMREF receives 0.4 million barrels per day of Arabian Light Crude oil (API gravity = 33, sulphur content = 1.75%)[65] while YASREF receives 0.4 million barrels per day of Arabian Heavy Crude oil (API gravity = 28, sulphur content = 2.9%)[66]. Since the SAMREF and YASREF refineries produce over 0.1 million barrels per day of diesel fuel, it was assumed that both refineries contribute an equal amount of Diesel fuel production reaching Makkah. Therefore, the refinery efficiency of 90.4% is calculated from GREET for Diesel using an average API gravity and sulphur content of 30.5 and 2.325%, respectively, for the input crude oil.

The refined products from Yanbu refineries are transported via pipeline to a Bulk storage plant located in North Jeddah over a distance of 303 km. The energy intensity of pipeline transport for Diesel was assumed to be the same for the crude oil at 260 BTU $ton^{-1}mile^{-1}$ [45] (Supplementary Note 9). The fuel is finally transported from the Bulk storage plant to retail fuel stations in Makkah via heavy-duty trucks over a distance of approximately 96 km. Therefore, the well-to-pump or the life-cycle energy use for Diesel fuel is calculated to be 166,466 BTU per mm BTU or 16.6% of the energy content of Diesel fuel dispensed at the pump. Based on the electricity grid in Saudi Arabia in 2022, the life-cycle $CO_2$ emission is calculated as 11,178 g $CO_2$ per mm BTU.

Among the four steps in the life-cycle of Diesel fuel, the energy use in Refinery is the largest. Therefore it was desirable to understand expected changes in future crude oil quality and its impact on refinery efficiency. Based on the OPEC World Oil Outlook[67], the API gravity and sulphur content for OPEC crude oil (a surrogate for Saudi crude oil) undergo negligible change over the ten-year period from 2020 to 2030 (see Supplementary Note 5). Therefore, in the near future, the only major change in the Diesel fuel cycle is expected to be in terms of lower $CO_2$ emissions as the Saudi electricity grid gets decarbonised in the coming years.

**Vehicle cycle**

**Vehicle specifications.** The specifications for the three types of buses can be found in the Supplementary Table 1.

While the primary distinction among different bus types lies in their powertrain and energy storage systems, both vehicle types share common body components such as doors, windows, seats, instrument panels, and controls, as well as similarities in chassis components, including brakes, suspension, wheels, and tyres, and bumpers[33,68,69].

The construction of an FCBs predominantly relies on the component ratios of BEBs, with adjustments made for the powertrain and storage system. Given the Toyota Sora's total weight of 12,464 kg, it was assumed that the material distribution, excluding the powertrain, mirrors that of BEBs. For propulsion purposes, the Sora is equipped with four battery modules (each with a 6.5 Ah capacity), rather than for emergency power supply. Prismatic NiMH battery modules from Panasonic, each comprising six 1.2V cells in series, are utilised by Toyota[70]. The module features a rated voltage of 7.2V, a 6.5 Ah capacity, a weight of 1.04 kg, and dimensions of 19.6 mm (W) × 106 mm (H) × 275 mm (L)[70]. This yields an estimated total battery weight of 214 kg for the Toyota Sora. Concerning the PEM FC stack, the Toyota Mirai's PEM FC stack (114 kW) weighed 56 kg[70]. Considering the Sora was fitted with two PEM FCs, it was inferred that the Sora's PEM FC stack weighs 112 kg[71]. Regarding high-pressure hydrogen tanks, the Sora is equipped with 10 tanks rated at 70 MPa. Given a reported combined weight of approximately 87.5 kg for two high-pressure hydrogen tanks, the estimated weight for ten hydrogen tanks was 437.5 kg[72]. Furthermore, following the GREET's life-cycle coefficient of the FC stack and auxiliary system data which offers a weight-to-power ratio of 1.27kg $kW^{-1}$, the weight of the FC balance-of-plant can be projected to be 179 kg[33].

**Replacement information.** Lead-acid batteries, commonly used in ICEBs, typically have a service life of ~3–4 years[73]. In contrast, NiMH batteries can sustain around 3000 cycles, while LFP batteries, used in BEBs, have a capacity of up to 6000 cycles[74].

The three bus types differ in their battery configurations: ICEBs incorporate lead-acid batteries, BEBs use LFP batteries, and FCBs are equipped with 6.5 Ah NiMH batteries. Assuming BEBs undergo two charging cycles daily, the LFP batteries are expected to last for roughly 16 years. PEM fuel cell stacks have an average operational lifespan of 50,000 to 80,000 hours, exceeding the bus lifespan considered in this study[75]. Hence, within a 10-year horizon, neither BEBs nor FCBs are anticipated to require LFP battery or PEM fuel cell stack replacements. In contrast, NiMH batteries in ICEBs will likely need one replacement, and lead-acid batteries could require three replacements over their operational lifespan.

For further details on the replacement data for other bus components, please refer to Supplementary Table 3.

**Recycled materials used in buses.** Considering the majority of buses in operation within Saudi Arabia were imported, the study posited that ICEBs originate from Sweden, BEBs from China, and FCBs from Japan. As a result, production-related data were assumed to be country-specific.

The estimated shares of recycled steel in vehicles for Japan, China, and Sweden were respectively 23.2%, 10.1% and 41.6%[35]. In addition, the shares of recycled aluminium in these countries were assumed to be 99.4%, 19.2%, and 70.9%, respectively[36]. In the 2030 scenario, all three types of buses, manufactured in Saudi Arabia, will utilise data from the GREET model.

**Shipping.** To estimate the emissions and the energy consumption during the shipping stage, a model was developed that utilises energy consumption and emission estimation equations, based on the International Council on Clean Transportation method, as outlined in equation (1) and equation (2) respectively[76].

$$\text{FC} = \sum_{t=0}^{t=n} \left( \left( P_{\text{ME}i} \times \text{LF}_{i,t} + D_{\text{AE}p,i,t} + D_{\text{BO}p,i,t} \right) \right) \times \text{SF}_{m,i} \qquad (1)$$

$$E_{i,j} = \sum_{t=0}^{t=n} \left( \left( P_{\text{ME}i} \times \text{LF}_{i,t} + D_{\text{AE}p,i,t} + D_{\text{BO}p,i,t} \right) \right) \times \text{EF}_{j,m} \qquad (2)$$

$i$ = Ship type
$j$ = Pollutant
$m$ = Fuel type
$p$ = Phase (cruise, manoeuvering, anchor, berth)
$t$ = Time (operating hour, h)
FC = Total fuel consumption
$\text{SF}_{m,i}$ = Specific fuel consumption for ship $i$, and fuel $m$
$E_{i,j}$ = Emissions (g) for ship $i$ and pollutant $j$
$P_{\text{ME}i}$ = Main engine power (kW) for ship $i$
$\text{LF}_{i,t}$ = Main engine load factor for ship $i$ at time $t$
$\text{EF}_{j,m}$ = Engine emission factor (g kWh$^{-1}$) for pollutant $j$ and fuel $m$
$D_{\text{AE}_{p,i,t}}$ = Auxiliary engine power demand (kW) in phase $p$ for ship $i$ at time $t$
$D_{\text{BO}_{p,i,t}}$ = Boiler power demand (kW) in phase $p$ for ship $i$ at time $t$

Furthermore, the cruising time for a vessel transporting buses can be determined using equation (3).

$$t = \frac{\text{Crusing distance}}{\text{The average speed of a ship}} \qquad (3)$$

The shipping emissions stem from the functioning of main engines, auxiliary engines, and boilers[76]. The specific fuel consumption and engine power were 217 g kWh$^{-1}$ and 2360 kW per cylinder, respectively[77–79]. The ship was estimated to have 8 cylinders[78], and an average vessel speed of 19.8 knots was assumed[77]. Additionally, based on the International Maritime Organisation's data, the roll-on/roll-off vessel's average auxiliary engine power and boiler power were assumed to be 1518 kW and 225 kW, respectively[80]. The roll-on/roll-off vessel's dead weight at the design draft amounted to 26,700 tons[77]. Heavy fuel oil's lower heating value is estimated at 39 MJ kg$^{-1}$ [81]. For further details on the emission factors, refer to Supplementary Table 2.

**Bus end-of-life.** In the end-of-life stage, aluminium and steel are the predominant materials recycled within the automotive industry. Considering the recycling context in Saudi Arabia, this study established that by 2022, the recycling rates for aluminium and steel are 33.9% and 10%, respectively[82–84]. For BEBs, it is essential to recycle LFP batteries. Currently, Saudi Arabia does not have battery recycling facilities[85]; however, the United Arab Emirates intends to construct one in the future. As a result, this research posits that by 2030, Saudi Arabia could have a battery recycling plant, facilitating the recycling of 30% of LiCO$_3$ in batteries via hydrometallurgy within the 2030 scenario. Furthermore, this investigation projects that advancements in aluminium and steel recycling technology for buses in Saudi Arabia will lead to recycling rates of 50% for aluminium and 20% for steel in the 2030 scenario.

## Bus energy consumption during operation stage

This study investigated the energy usage of a bus's powertrain and AC system during the operation stage. To thoroughly investigate the additional life-cycle environmental burden imposed by heating and cooling, an integrated model to calculate the cooling load of the AC system was developed. The life-cycle coefficients of fuels were sourced from the well-established GREET model. Given the typical annual outdoor temperature in Makkah of 39 °C, the cooling load escalates to a substantial 32.2 kW in line with previous research[86]. In pursuit of a just evaluation across three distinct bus types, this study operates under the assumption that each bus carries 30 passengers and that the surface areas of the buses are approximately analogous. This enhanced method facilitated a comparison of cooling load results between bus operations in Saudi Arabia and those in other regions, addressing potential limitations present in the existing literature.

The total cooling load, $\dot{Q}_{\text{total}}$, for the AC system within a bus cabin arises from a variety of heat sources, as shown in equation (4). These encompass the metabolic heat, $\dot{Q}_{\text{met}}$, produced by passengers and drivers, the solar heat load, $\dot{Q}_{\text{sun}}$, originating from direct, diffuse, and reflected radiation, heat emissions from ventilation systems, and heat exchange, $\dot{Q}_{\text{amb}}$, resulting from temperature gradients between the bus structure—including windows, walls, floor, roof, and doors—and the external environment. Moreover, the heat generated by the vehicle heating system, $\dot{Q}_{\text{hsyst}}$, should be considered to ensure passenger thermal comfort[87]. Notably, this study did not encompass the heat loads generated by distinct powertrain components, $\dot{Q}_{\text{uni}}$, of three bus types: the fuel cell stack for the PEM FCBs, the electric motor and LFP battery for the BEBs, and the heat produced by the internal combustion engine (ICE) and exhaust heat for the ICEBs.

$$\dot{Q}_{\text{ACtotal}} = \dot{Q}_{\text{met}} + \dot{Q}_{\text{sun}} + \dot{Q}_{\text{ven}} + \dot{Q}_{\text{amb}} + \dot{Q}_{\text{hsyst}} + \dot{Q}_{\text{uni}} \qquad (4)$$

The quantified heat load for the AC system, combined with baseline energy consumption, determines the total operational energy demand of the bus. The cooling load from the ICE and exhaust gas can be considered negligible due to the effective insulation of most vehicles[88,89]. For BEBs, a variety of battery thermal management systems are employed, including passive cabin cooling using air, active moderate liquid circulation using refrigerant, and active liquid circulation using refrigerant and coolant[90,91]. In this study, it was assumed that BEBs utilise either active moderate liquid cycling or active liquid cycling battery thermal management systems, resulting in minimal impact on the cooling load of the battery and motor. In the case of FCBs, various cooling methods can be applied, such as heat spreaders, separate air flow, liquid cooling, and phase change methods[92]. Among these, liquid cooling has been considered the most suitable technology for vehicle applications due to its high specific heat capacity and ease of integration with cooling systems[92–94]. The cooling/heating system for the FC stack is typically incorporated into the balance-of-plant components[95,96]. Therefore, for the purpose of this analysis, the cooling load of the AC system in PEM FC buses was assumed to be independent of the FC stack.

Passenger satisfaction and the overall transportation experience are strongly influenced by the optimisation of comfort temperature within a vehicle's cabin. This study sets the comfort temperature at 25 °C, taking into account occupants wearing short-sleeved shirts and long pants, which corresponds to approximately 0.5 clo under summer conditions[97]. In line with the ISO 8996 standard, metabolic heat production rates, denoted as $M$, are presumed to be 55 W m$^{-2}$ for passengers and 85 W m$^{-2}$ for the bus driver[98]. The average human body's surface area, represented as $A_{\text{Du}}$, is approximated at 1.8 m$^2$ [99]. For a bus accommodating 30 seated passengers, the metabolic load ($\dot{Q}_{\text{Met}}$) is derived as 3.12 kW, as determined by Eq. (5). The Global Solar Atlas provides data on the direct normal irradiance in Makkah, indicating a value of 2239 kWh m$^{-2}$ [100]. The study's geographic focus is Makkah, characterised by latitude $\varphi$ (21.3891°N) and longitude $\lambda$ (39.8579°E). The bus surface tilt angles, symbolised by $\sum$, are established as

90° for vertical walls and 0° for the roof.

$$\dot{Q}_{\text{Met}} = \sum_{\text{Passengers}} M A_{\text{Du}} \tag{5}$$

The dynamics of Earth's solar environment play a crucial role in understanding and optimising energy systems. One such essential aspect is the solar declination angle ($\delta$), which defines the angular relationship between the Earth-Sun line and the equatorial plane. This angle experiences annual cyclic variations due to the Earth's equatorial plane's 23.45° inclination relative to its orbital plane. The annual changes in $\delta$ are calculated using Equation (6), where 'n' indicates the day of the year[101]. The apparent solar time (AST) equation is referenced in the ASHRAE Handbook[97].

Furthermore, the hour angle ($\omega$) signifies the sun's eastward or westward angular displacement from the local meridian, affected by Earth's rotation[97]. The solar altitude angle $\beta$ denotes the angle between the horizontal plane and a line extending from the sun. This angle is 0° when the sun is on the horizon and 90° when the sun is directly overhead[97]. The solar azimuth angle ($\phi$), which represents the southward angular deviation originating from the Earth-Sun line's projection onto a horizontal surface, also warrants consideration in this paper[97]. The study assumed that the bus surface azimuth $\gamma$ and solar azimuth $\phi$ values are closely related. The relationships between these angles are presented in Equations (6) through (10):

$$\delta = 23.45 \cdot \sin\left(\frac{360°}{365} \cdot (n + 284)\right) \tag{6}$$

$$\text{AST} = \text{SAST} - 2\,\text{h} + \frac{\lambda}{15°/h} \tag{7}$$

$$\omega = \frac{360°}{24h}(\text{AST} - 12) \tag{8}$$

$$\beta = \arcsin\left(\sin\varphi \cdot \sin\delta + \cos\varphi \cdot \cos\delta \cdot \cos\omega\right) \tag{9}$$

$$\phi = \frac{\sin\beta \sin\varphi - \sin\delta}{\cos\beta \cos\varphi} \tag{10}$$

Energy-efficient transportation systems demand a comprehensive understanding of the effects of solar radiation on a vehicle's cabin environment under diverse weather conditions. This study delved into buses operating in Saudi Arabia, a region characterised by predominantly clear skies. The total irradiance ($I_t$), as defined in equation (11), received by the surface under clear skies comprises three distinct components: direct beam from the sun ($I_D$), diffuse reflection from the entire sky dome ($I_d$), and ground reflection from the ground in front of the receiving surface ($I_{\text{ref,S}}$)[97,102]. The calculations for these factors are outlined from equation (12) to equation (15).

Within the context of buses, the investigation of diffuse reflection is confined to vertical and horizontal surfaces. Accordingly, the shape factor ($F_{\text{ss}}$) assumed values of 0.5 for vertical surfaces and 1 for horizontal surfaces[102]. Equation (14) offers a means to compute corresponding values for surfaces with varying tilt angles[102]. The methodologies utilised for assessing heat absorption via windows and heat exchange between the bus exterior and its environment follow established literature guidelines[102].

A critical parameter in this investigation is the sol-air temperature ($T_{\text{sol}}$), as defined in equation (16). This parameter accounts for the combined influence of solar radiation on the bus's exterior surface and the inward heat transfer due to the temperature differential between the ambient and cabin environments[97]. The absorptance ($\alpha$) and heat transfer coefficient ($h_o$) ratio was assumed to be 0.026 for light-coloured surfaces[103]. For vertical surfaces, the product of hemispherical emissivity ($\varepsilon$) and the accurate value ($\triangle R$) was assumed to be zero, as suggested by the ASHRAE handbook[97]. For horizontal surfaces, which receive long-wave radiation solely from the sky, a suitable value of $\triangle R$ is estimated at 63 W m$^{-2}$[97].

Considering the implications of air movement on heat transfer dynamics, average convection coefficients $h_o$, as delineated in equation (17)[99], are calculated for the bus's external surfaces using an average velocity ($v$) of 34.8 km h$^{-1}$, which yields a value of 25 W m$^{-2}$ K$^{-1}$. Consequently, the long-wave correction term is established at a value of $-15$ °C.

$$I_t = I_D + I_d + I_{\text{ref,S}} \tag{11}$$

$$I_D = I_{\sum} = I_{\text{DN}} \cdot \cos\theta_{\sum} = I_{\text{DN}}\left(\cos\beta \cos\gamma \cos{\textstyle\sum} + \sin\beta \cos{\textstyle\sum}\right) \tag{12}$$

$$I_d = \frac{C I_{\text{DN}} F_{\text{ss}}}{C_n^2} \tag{13}$$

$$F_{\text{ss}} = \frac{1.0 + \cos\sum}{2} \tag{14}$$

$$I_{\text{ref}} = \rho_s F_{\text{sr}}(I_D + I_d) \tag{15}$$

$$T_{\text{sol}} = T_o + \frac{\alpha I_t}{h_o} - \frac{\varepsilon \triangle R}{h_o} \tag{16}$$

$$h_o = 9 + 3.5 v^{0.66} \tag{17}$$

In pursuit of a deeper understanding of heat load in energy-efficient transportation systems, the present investigation categorised heat load into three primary components: heat transfer through the window ($\dot{Q}_{\text{glass}}$), heat transfer through the bus wall ($\dot{Q}_{\text{body}}$), and heat transfer through other parts of the bus ($\dot{Q}_{\text{others}}$). These components are elucidated in equation (18). A critical aspect of investigating heat transfer through windows is the solar heat gain factor (SHGF), representing the average solar heat gain during cloudless days in W m$^{-2}$[102]. Considering the geographical proximity between Iran and Saudi Arabia, this study adopted an average SHGF value of 172.5 W m$^{-2}$ for bus glass in Saudi Arabia[104]. Additionally, the study assumed a shading coefficient (SC) of 0.811[105]. Details of bus element surface areas ($A$), heat transfer coefficient ($K$), and solar collector (SC) parameters are provided in Supplementary Table 4. Window and other structural areas were assumed to be consistent across all three bus models[105].

$$\dot{Q}_{\text{sun}} + \dot{Q}_{\text{amb}} = \dot{Q}_{\text{glass}} + \dot{Q}_{\text{body}} + \dot{Q}_{\text{door}} \tag{18}$$

To assess heat transfer through windows, the analysis encompasses various glass components, including front, rear, and side windows, driver's windows, and skylights. The total heat transfer ($\dot{Q}_{\text{glass}}$) is determined as the sum of the individual heat transfer contributions—$\dot{Q}_{\text{gfront}}$, $\dot{Q}_{\text{grear}}$, $\dot{Q}_{\text{gside}}$, $\dot{Q}_{\text{driver,windows}}$, and $\dot{Q}_{\text{skylight}}$—as expressed in equations (19) to (25)[105].

$$\dot{Q}_{\text{glass}} = \dot{Q}_{\text{gfront}} + \dot{Q}_{\text{grear}} + \dot{Q}_{\text{gside}} + \dot{Q}_{\text{driver,windows}} + + \dot{Q}_{\text{skylight}} \tag{19}$$

$$\dot{Q}_{\text{gfront}} = K_{\text{gf}} A_{\text{gf}}(T_o - T_i) + A_{\text{gf}} \text{SHGF}_{\text{max}} * \text{SC} \tag{20}$$

$$\dot{Q}_{\text{grear}} = K_{\text{gr}} A_{\text{gr}}(T_o - T_i) + A_{\text{gr}} \text{SHGF}_{\text{max}} * \text{SC} \tag{21}$$

$$\dot{Q}_{\text{gside}} = K_{\text{gs}} A_{\text{gs}}(T_o - T_i) + A_{\text{gs}} \text{SHGF}_{\text{max}} * \text{SC} \tag{22}$$

$$\dot{Q}_{\text{driver,windows}} = K_{\text{dw}} A_{\text{dw}}(T_o - T_i) + A_{\text{dw}} \text{SHGF}_{\text{max}} * \text{SC} \tag{23}$$

$$\dot{Q}_{\text{door}} = K_d A_d(T_o - T_i) + A_d \text{SHGF}_{\text{max}} * \text{SC} \tag{24}$$

$$\dot{Q}_{\text{skylight}} = K_{\text{sl}} A_{\text{sl}}(T_o - T_i) + A_{\text{sl}} \text{SHGF}_{\text{max}} * \text{SC} \tag{25}$$

In order to assess the heat transfer through the bus wall and roof, the cooling load of the cabin, attributed to heat transfer through the bus walls, is defined by equations (26) to (31). $T_o$ and $T_i$ indicate the outdoor temperature and the cabin temperature, respectively. It was noteworthy that the heat transfer through the door was assumed to be negligible, given that the bus will not stop ($t_{stop} = 0h$) during the running time ($t_{run} = 2h$), as delineated in equation (32)[87].

$$\dot{Q}_{body} = \dot{Q}_{roof} + \dot{Q}_{floor} + \dot{Q}_{side,panel} + \dot{Q}_{b,front} + \dot{Q}_{b,rear} \tag{26}$$

$$\dot{Q}_{roof} = K_r A_r (T_{ot} - T_i) + h_o A_r (T_{sol} - T_o) \tag{27}$$

$$\dot{Q}_{floor} = K_f A_f (T_o - T_i) \tag{28}$$

$$\dot{Q}_{side,panel} = K_{sp} A_{sp} (T_o - T_i) + h_o A_{sp} (T_{sol} - T_o) \tag{29}$$

$$\dot{Q}_{b,front} = K_{bf} A_{bf} (T_o - T_i) + h_o A_{bf} (T_{sol} - T_o) \tag{30}$$

$$\dot{Q}_{b,rear} = K_{br} A_{br} (T_o - T_i) + h_o A_{br} (T_{sol} - T_o) \tag{31}$$

$$Q_{door} = V \rho_{air} c \Delta T \frac{t_{stop}}{t_{run}} \tag{32}$$

A factor in evaluating the efficiency of bus cooling systems is the coefficient of performance (COP) of the compressor. With a hypothesised COP value of 1.6[106], the relationship between cooling capacity and compressor power can be described through equation (33).

$$P_{cp} = \frac{Q_{ACtotal}}{COP} \tag{33}$$

The energy consumption of FCBs is assumed to be 5.5 kg $H_2$ per 100 km, excluding AC and auxiliary systems[37,107]. The operating temperature range of the PEM FC stack falls between 60 °C and 80 °C, considerably higher than the average outdoor temperature in Makkah[108–111]. Thus, it was postulated that the PEM FC stack's operation would remain unaffected by outdoor temperature, implying that the baseline energy consumption would similarly remain consistent regardless of external temperature.

BEBs have been identified to consume 0.9 kWh km$^{-1}$ under the same conditions[37,112,113]. Extensive studies suggest that a temperature range of 25 °C to 40 °C is deemed optimal for charging and discharging LFP batteries[114–119]. Consequently, it was postulated that an ambient temperature of 39 °C would not influence the baseline performance of BEBs.

For buses powered by ICE, energy consumption without AC system has been estimated to be 30 $l_{diesel}$ per 100 km[120–124].

The energy conversion efficiencies across pump-to-wheel stages for FCBs, BEBs, and ICEBs are detailed in Supplementary Table 5. Supplementary Table 6, provides data on the total tank-to-wheel energy consumption for each bus type.

### 2030 scenarios

In order to investigate the potential changes in LCA results in the near future, specifically by 2030, we developed a comprehensive scenario known as the "2030 scenario". The scenario includes various factors such as an increased renewable energy share in Saudi Arabia, advancements in CCS technology efficiency, shale gas utilisation, vehicle light-weighting strategies, localised vehicle production within Saudi Arabia (with implications on the ADR, battery recycling, and shipping phases), and projected enhancements in FC stack efficiency. The specific details of these factors are elaborated in Table 1.

### Renewable energy-powered BEBs: scenarios for 2022 and 2030.
This study evaluates the carbon reduction potential of BEBs powered solely by renewable energy, juxtaposed with a scenario of 100% blue hydrogen, for the years 2022 and 2030. It assumes that all electricity from the "tank-to-wheel" phase is sourced from solar energy. However, the environmental impact of photovoltaic plant infrastructure is not considered within this analysis. For both scenarios, the electricity consumed in the vehicle manufacturing phase and RFS infrastructure construction is drawn from the grid.

### 2030 mixed hydrogen scenario.
For a balanced assessment against grid-connected BEBs in 2030, this study delves into a scenario envisioning FCBs powered by a hybrid hydrogen mix. According to projections, this mix is anticipated to comprise 98% blue and 2% grey hydrogen[42].

### Data availability
The LCIA data, results, and scenarios information are publicly accessible within this paper and Supplementary Information. For any further data that supports the findings of this study, interested individuals can request access from the authors.

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

## Acknowledgements
This work was supported by the KAUST FLEET consortium and its member companies. The authors express their gratitude to Qi Wang for insightful suggestions on manuscript revisions.

## Author contributions
M.S. acquired the funding and directed the project. M.S., C.Z., and L.K. designed the research plan. C.Z. and L.K. compiled the life-cycle inventories and conducted the life-cycle assessment for the vehicle, hydrogen, and electricity cycles. A.A. worked on the life-cycle inventories and conducted the LCA for the diesel cycle. C.Z. and A.A. prepared the figures and tables and compiled the Supplementary material. J.M. and E.C. provided guidance on the fuel cycle, while N.M. provided guidance on the bus fleet and bus route. M.S., C.Z., L.K., and A.A. collaboratively analyzed the data and wrote the paper. All authors participated in the discussion and preparation of the paper.

## Competing interests
