## [Peer Review File · Communications Engineering]

Reviewers' comments:

Reviewer #1 (Remarks to the Author):

Dear Editor and Authors,

Thank you for the opportunity to review the manuscript titled "Solutions for Decarbonizing Urban Bus Transport in non-OECD Countries: A Life Cycle Case Study in Saudi Arabia" by Professor Sarathy and co-workers. The authors presented a study about decarbonization potential of advanced vehicle technologies, analysis of emissions and energy utilisation of FCBs, BEBs, and internal combustion engine buses (ICEBs) across their entire life cycle, including the energy and emissions related to bus transportation. In addition, the study introduces an LCA model for bus systems, based on conditions outside the OECD but adaptable universally. The topic of this article is appropriate for the scope of "Communications Engineering," and the research methods used are sufficiently comprehensive. This article is interesting, well-written and contains a lot of novelty

There are some issues [major revision] that authors must resolve before a manuscript will be published. The following are my comments.

1. The title does not correspond with the abstract and introduction, which focus on heavy-duty transportation rather than public transportation via buses. Both the abstract and the introduction must be supplemented with this information if the title is to remain as it is.
2. Methodology should be found before the Results
3. Information about fuel/electricity consumption should be expressed in basic units and not in MJ/km
4. The "functional unit" should be clearly defined.
5. The article lacks references to relevant works including the following:
 - Bouter, A., Hache, E., Ternel, C., Beauchet, S., 2020. Comparative environmental life cycle assessment of several powertrain types for cars and buses in France for two driving cycles: "worldwide harmonized light vehicle test procedure" cycle and urban cycle. *Int. J. Life Cycle Assess.* 25, 1545–1565. <https://doi.org/10.1007/s11367-020-01756-2>.
 - Nordelöf, A., Romare, M., Tivander, J., 2019. Life cycle assessment of city buses powered by electricity, hydrogenated vegetable oil or diesel. *Transp. Res.D Transp. Environ.* 75, 211–222. <https://doi.org/10.1016/j.trd.2019.08.019>.
6. The descriptions of the drawings (fig. 5 and fig. 6) are too extensive and should be included in the text

I look forward to hearing from you.

Regards,

Reviewer

Reviewer #2 (Remarks to the Author):

What are the major claims of the paper?

The paper analyses the life cycle impacts of busses using different powertrains and fuels in Saudi-Arabia, focussing on energy consumption and greenhouse gas (GHG) emissions.

It finds that, at present, fuel cell electric busses reduce emissions most compared to internal combustion engine busses, albeit causing higher energy consumption. Battery electric vehicles are found to perform better in terms of energy consumption, but to reduce emissions to a lesser extent. For 2030 conditions, battery electric busses catch up to some extent.

Are they novel and will they be of interest to others in the community and the wider field? If the conclusions are not original, it would be helpful if you could provide relevant references.

- The finding that BEV busses perform worse, in terms of GHG emissions, than FCEV busses in the use phase (fuel cycle) is somewhat surprising, in many studies I have seen (often focussing on passenger cars and Europe though) the known pattern is that BEV perform better in the use phase and therefore break even after a few years of use. However, this finding depends on the assumptions made for the electricity mix and pathways for hydrogen production selected, which I fear are somewhat biased, see also concerns raised below.

- It seems that the method as such is not novel, but extended to more detail for busses compared to previous studies, albeit based on simple extrapolation of weight-based patterns from cars. It is not discussed to what extent this is realistic.

Is the work convincing, and if not, what further evidence would be required to strengthen the conclusions?

I am missing detail on the assumptions made for being able to fully judge the validity of conclusions. Main items listed in the concerns below.

On a more subjective note, do you feel that the paper will influence thinking in the field?

Personally I am not sure it will. The fact that the choice of fuel/energy pathways is decisive for the life cycle balance of different vehicle fuel and powertrain options is well known. If the authors could substantiate better the choice of their pathways and prove the case that for H2, better production pathways are and remain available than for electricity, this could be of interest. However, it is mentioned several times in the paper that the Saudi-Arabian context is rich in sun, therefore a carbon-intensive production of electricity seems to be a choice, not a given.

Please feel free to raise any further questions and concerns about the paper.

- Main assumptions on operation conditions and bus specifications are not specified. This regards, among others, the drive profiles and mileage assumptions used, bus size and motorization, battery and fuel tank capacities etc., as well an explanation under what conditions calculations apply (e.g., real world operation, or some cycles).

- I just saw one hint in figure 1 that the busses run two trips of 70km per day and are used for 4 hours a day. This assumption seems to be very low. Given the investment cost of busses I would expect an economically viable activity of at least double the trips per day.

- If busses are to do 70km trips with a one hour stop at a given depot in-between, battery capacity (as well as H2 tank size) can be quite low. Intuitively, I would expect the BEV battery to cause less production burden in this case.

- Also the time frame of vehicle use considered is not specified. I would have expected that BEV fare better, in particular over long use periods and large lifetime activity, given their superior energy efficiency. Only kWh/km are specified in the paper so it is hard to check.

- The fact that urban busses have specific operation profiles is not discussed. For example, what is assumed with regard to refuelling/recharging/infrastructure? It seems that the build up of dedicated infrastructure is included in the calculations, but is there any reflection of the fact that urban busses

typically have standardized routes and can be charged in a dedicated depot, thus not necessarily requiring the build-up of wide-spread infrastructure?

- It seems that, for hydrogen, two rather advanced production routes (including also CCS, which to my information is not yet a widely applied standard technology) are used. In contrast, for electricity, the paper uses grid mix. This seems a somewhat biased assumption. Either electricity grid mix needs to be compared to average today H2 production pathway, or for both ambitious assumptions should be made for a fair comparison (e.g., marginal electricity production for BEV, potentially taking into account the option of providing dedicated renewable energy plants for producing electricity close to depot).

- The energy aspect is not discussed appropriately. Figure 5 shows that the lease GHG intense option (blue H2 FCEV) consumes about 20% more energy than grey H2 FCEV or BEV in 2022, and the gap versus BEV grows to almost 40% by 2030. This could raise some concerns over energy availability and affordability of the preferred solution. Moreover, in the conclusions, efuels are mentioned (without any evidence or analysis) as a promising option to decarbonize ICE, whereas it is known from several studies that efuel and ICE are both very energy inefficient options. Energy efficiency should be better discussed in the paper.

- In Table 1, for 2022, the share of recycled steel and aluminium is different for the three bus types (while in 2030 it is the same). Why is this the case? I expect these to be the materials used for the vehicle body, in that case shouldn't it be similar?

- Figure axes labels come out very small (particularly figure 3).

We would also be grateful if you could comment on the appropriateness and validity of any statistical analysis, as well the ability of a researcher to reproduce the work, given the level of detail provided.

Main assumptions are not specified, see previous concerns. To my view, the analysis is not reproducible with the data provided.

Ms. Ref. No.: COMMSENG-23-0358A

Title: Solutions for Decarbonizing Urban Bus Transport in non-OECD Countries: A Life Cycle Case Study in Saudi Arabia

Detailed Response to Reviewers

The authors would like to thank the editor and the reviewers for taking the time to review our manuscript titled “Solutions for Decarbonizing Urban Bus Transport in non-OECD Countries: A Life Cycle Case Study in Saudi Arabia”. We are sincerely grateful for the comments and suggestions provided by the reviewers and editors, which have greatly aided in improving the quality of our paper. In response to this constructive feedback, we embarked on comprehensive revisions and enhancements of our manuscript. Specifically, we executed a strategic restructuring of the manuscript, seamlessly integrating substantial portions of material from the Supplementary Information into the main text, in addition to incorporating detailed elaborations and novel experimental findings. A principal aspect of our revision concentrated on elucidating the functional unit and introducing additional experiments on the sensitivity analysis regarding the bus capacity utilization factor, with the objective of evaluating their impact on the diverse buses scrutinized in this investigation. These supplementary data not only address the issues concerning the functional unit and passenger count but also provide a thorough analysis of hydrogen production methodologies. Moreover, we accorded particular consideration to the feedback from Reviewer #2, especially with regard to shedding light on the innovative elements of our study and offering an exhaustive explanation of our assumptions. Through the strategic reallocation and augmentation of content related to methodology within the main text, we have significantly advanced the study’s reproducibility, clarity, and its applicability to a broader context, transcending the confines of the Saudi scenario. We are convinced that these modifications have substantially strengthened the manuscript’s foundation and effectively addressed the concerns delineated by the reviewers.

(Additionally, we have addressed feedback from the editor, detailed within the Author Cover Letter.)

Here are our responses to the reviewers' comments:

Reviewers' comments:

Reviewer #1 (Remarks to the Author):

Dear Editor and Authors,

The following are my comments.

1. The title does not correspond with the abstract and introduction, which focus on heavy-duty transportation rather than public transportation via buses. Both the abstract and the introduction must be supplemented with this information if the title is to remain as it is.

We have changed the introduction, and abstract accordingly.

Abstract: "With the vast majority of the global population, non-OECD countries such as Saudi Arabia, are facing a harder battle against carbon emissions. Saudi Arabia's urban bus transportation is under pressure to decarbonize, and new energy vehicles can mitigate emissions. This life-cycle assessment illustrates that, despite an energy consumption surge of 19.7% due to carbon capture and storage (CCS), the fuel cell buses (FCBs) powered by blue hydrogen can diminish carbon emissions by 53.5% versus internal combustion engine buses. The battery electric buses (BEBs), affected by Saudi's energy mix and Lithium-ion batteries manufacturing, only cut emissions by 16.7%. These climate benefits pivot significantly on energy sources, and the efficiencies of CCS and hydrogen production. With projected technological advancements and 50% renewable energy by 2030, FCBs and BEBs could achieve further carbon reduction by 19.3%-51%. However, FCBs consistently demonstrate lower environmental burdens, while BEBs manifest the highest impacts."

Introduction (lines 50-55): "GHG emissions stemming from transportation in Saudi Arabia account for 22% [7,8]. In line with Vision 2030 and the Sustainable Development Goals (SDGs), Saudi Arabia confronts the urgent task of diminishing dependence on private vehicles while enhancing urban bus transportation. Integrating innovative vehicle technologies to decarbonize public transport sector is of paramount importance for Saudi Arabia."

2. Methodology should be found before the Results

Thank you for your comments regarding our manuscript's structure. We adhered to the standard format prevalent in the Communication Engineering journal, which requires us to place the Methods section after the Results and Conclusions.

3. Information about fuel/electricity consumption should be expressed in basic units and not in MJ/km

Thank you for your inquiry. In response to your question regarding the use of functional units in LCA studies, it is indeed common practice to employ consistent functional units when assessing both energy consumption and emissions. Typically, this functional unit is represented as "1 kilometer traveled by bus," and the results are presented in terms of energy (measured in MJ/km).

This approach aligns with methodologies documented in relevant literature. For specific examples and detailed methodologies, you may refer to the studies cited as 'Reference (Iannuzzi, Hilbert, & Lora, 2021)' and 'Reference (Lucas, Neto, & Silva, 2012).'

Additionally, we have included the results in the basis unit in the supplementary materials of our study. These results provide further insights into our findings and methodologies. Please refer to the Supplementary Note 8.

4. The "functional unit " should be clearly defined.

Thank you to the reviewer for pointing out that the "functional unit" should be clearly defined. We added this definition to the manuscript on lines 484-495, page 19.

"4.1 Functional unit

In the framework of LCA, the "functional unit" is a crucial metric that quantifies the functionality of the products under investigation. Specifically, in evaluating bus systems, they are classified into two cycles: fuel cycle and vehicle cycle. To methodically assess energy consumption and emissions, the functional unit is defined as "1 kilometer traveled by a bus" under general operational conditions. Public transport systems exhibit passenger number fluctuations due to temporal and seasonal factors. To mitigate the impact of such variations, this investigation employs a hypothetical scenario where three bus models, each with a capacity for 30 passengers, operate under uniform load conditions. This approach ensures an equitable comparison of life cycle emissions across the bus models, effectively neutralizing the discrepancies arising from dynamic passenger numbers."

5. The article lacks references to relevant works including the following:

- Bouter, A., Hache, E., Ternel, C., Beauchet, S., 2020. Comparative environmental life cycle assessment of several powertrain types for cars and buses in France for two driving cycles: "worldwide harmonized light vehicle test procedure" cycle and urban cycle. *Int. J. Life Cycle Assess.* 25, 1545–1565. <https://doi.org/10.1007/s11367-020-01756-2>.
- Nordelöf, A., Romare, M., Tivander, J., 2019. Life cycle assessment of city buses powered by electricity, hydrogenated vegetable oil or diesel. *Transp. Res.D Transp. Environ.* 75, 211–222. <https://doi.org/10.1016/j.trd.2019.08.019>.

Thank you for your valuable suggestion. In accordance with it, we have updated our manuscript to include the following literature references, specifically in the section spanning lines 56-62, page 3:

- Bouter, A., Hache, E., Ternel, C., Beauchet, S., 2020. Comparative environmental life cycle assessment of several powertrain types for cars and buses in France for two driving cycles: "worldwide harmonized light vehicle test procedure" cycle and urban cycle. *Int. J. Life Cycle Assess.* 25, 1545–1565. <https://doi.org/10.1007/s11367-020-01756-2>.

- Nordelöf, A., Romare, M., Tivander, J., 2019. Life cycle assessment of city buses powered by

electricity, hydrogenated vegetable oil or diesel. *Transp. Res.D Transp. Environ.* 75, 211–222. <https://doi.org/10.1016/j.trd.2019.08.019>.

• Lucas, A., Neto, R. C., & Silva, C. A. (2012). Impact of energy supply infrastructure in life cycle analysis of hydrogen and electric systems applied to the Portuguese transportation sector. *international journal of hydrogen energy*, 37(15), 10973-10985. <https://doi.org/10.1016/j.ijhydene.2012.04.127>.

- Lines 56-58: “Hydrogen fuel cell vehicles (FCVs) and battery electric vehicles (BEVs) are emission-free in the vehicle operation phase, offering a pathway to decarbonize the heavy-duty transport [9-25].”
- Lines 58-62: “However, the life-cycle environmental impacts of these vehicles are intrinsically linked to many other factors such as feedstock production, energy sources for electricity generation, fuel production efficiency, battery manufacturing, fuel cell (FC) stack performance, battery technology, and operational energy efficiency [9-11, 13, 15, 19, 21-27].”

6. The descriptions of the drawings (fig. 5 and fig. 6) are too extensive and should be included in the text.

Thank you for your suggestion regarding the descriptions of Figures 5 and 6. I've made an addition to our main text by including a new figure. As a result of this insertion, the numbering of the figures has been updated accordingly. What was previously Figure 5 is now renumbered as Figure 6, and the original Figure 6 has been changed to Figure 7.

We have taken the following steps to address your concerns:

- **Figure 6:** The detailed descriptions previously in the figure captions have been integrated into the manuscript.

Lines 300-303, page 11: “Figure 6 (a) illustrates the aggregated energy consumption per kilometer (MJ/km) for bus driving in 2022, while Figure 6 (b) graphically delineates the cumulative CO₂-eq emissions per kilometer (kg/km) for the same period.”

Lines 363-366, page 14: “Figure 6 (c) presents a detailed quantification of total energy consumption per kilometer (MJ/km) for bus operation in 2030. Concurrently, Figure 6 (d) provides an estimation of the total life-cycle CO₂-eq emissions per kilometer (kg/km) for bus driving projected for the same year.”

The revised caption for Figure 6 is: “Complete life-cycle results for bus energy expenditure and CO₂ emissions in 2022 and 2030: Incorporating all inclusive phases of both vehicle and fuel cycles. (a) Energy consumption (MJ/km) in 2022, (b) GHG emissions (kg/km) in 2022, (c) Energy consumption (MJ/km) in 2030, and (d) GHG emissions (kg/km) in 2030.”

- **Figure 7:** Similar adjustments have been made for Figure 7 in the manuscript.

Lines 393-399, page 14-15: "Figure 7 delineates the differential impacts elicited by parametric adjustments on the GWP100 for the year 2022, which elucidates the dichotomy between the exacerbation observed in a 'worst-case' scenario and the mitigation realized in a 'better-case' scenario, consequent to alterations in various parameters. Specifically, Figure 7 (a) to (d) expound upon the repercussions of these adjustments on distinct vehicle categories: blue FCBs, grey FCBs, BEBs, and ICEBs, respectively."

The updated caption for Figure 7 is: "GWP 100 comparative analysis for 2022: Contrasting 'worst-case' amplification and 'better-case' mitigation in blue and grey FCBs, BEBs, and ICEBs. Panels (a)-(d) show the $\pm 10\%$ factor impact on GWP100, with exceptions for different ranges. *RE adjustments based on a 50% renewable energy baseline."

Reviewer #2

What are the major claims of the paper?

The major claims of the paper titled "Solutions for Decarbonizing Urban Bus Transport in non-OECD Countries: A Life-Cycle Case Study in Saudi Arabia" are as follows:

- Decarbonization of Heavy-Duty Transportation in Saudi Arabia: The paper addresses the challenge of decarbonizing heavy-duty transportation in non-OECD countries, focusing on Saudi Arabia. It emphasizes the role of new energy vehicles, such as fuel cell buses (FCBs) and battery electric buses (BEBs), in mitigating emissions despite an increase in energy consumption due to carbon capture and storage (CCS).
- Comprehensive Life-Cycle Assessment (LCA): The study conducts a detailed life-cycle assessment of urban bus fleets in Makkah, Saudi Arabia. This assessment includes analyzing emissions and energy utilization of FCBs, BEBs, and internal combustion engine buses (ICEBs) across their entire life cycle. It also considers factors like bus transportation via Roll-on/Roll-off (RORO) vessels and the construction of charging and hydrogen refueling station (RFS) infrastructures.
- Comparison of Vehicle and Fuel Cycles for Different Bus Models: The paper provides insights into the energy consumption and greenhouse gas (GHG) emissions of different bus models during their vehicle and fuel cycles. It highlights that BEBs show the highest energy consumption and GHG emissions in the vehicle cycle, mainly due to the energy-intensive nature of lithium iron phosphate (LFP) battery manufacturing. The fuel cycle analysis includes the consideration of blue and grey hydrogen, electricity, and diesel as corresponding fuels for the different bus models, with a focus on methane leakage during natural gas production and its impact on GHG emissions.

- **Energy Consumption and Emissions in Different Scenarios:** The paper compares the 2022 and 2030 scenarios for bus operation, emphasizing the energy consumption and emissions of different bus models. It reveals that blue FCBs have the lowest life-cycle emissions in the 2022 scenario, followed by grey FCBs, BEBs, and ICEBs. The study notes the increase in energy consumption for blue FCBs due to the CCS process, while highlighting the lower energy consumption and emission reduction for BEBs.

The paper analyses the life cycle impacts of busses using different powertrains and fuels in Saudi Arabia, focussing on energy consumption and greenhouse gas (GHG) emissions. It finds that, at present, fuel cell electric busses reduce emissions most compared to internal combustion engine busses, albeit causing higher energy consumption. Battery electric vehicles are found to perform better in terms of energy consumption, but to reduce emissions to a lesser extent. For 2030 conditions, battery electric busses catch up to some extent.

Are they novel and will they be of interest to others in the community and the wider field? If the conclusions are not original, it would be helpful if you could provide relevant references.

The conclusions drawn in the paper are indeed of significant interest to the community and the wider field, particularly in the context of sustainable transportation and energy policy. The novelty and relevance of these findings can be evaluated based on several aspects:

- **Contextual Relevance:** The focus on Saudi Arabia, a non-OECD country with a unique energy landscape heavily reliant on fossil fuels, adds contextual novelty. Much of the existing research on decarbonizing transportation is often centered around OECD countries, where the energy mix and transportation challenges can be markedly different.
- **Comprehensive Life-Cycle Assessment:** The detailed life-cycle assessment (LCA) of different bus powertrains and fuels, encompassing energy consumption and GHG emissions, is a comprehensive approach. While LCAs are common in environmental studies, the specific application to urban bus fleets in Saudi Arabia provides new insights, particularly considering different powertrain technologies (FCBs, BEBs, ICEBs) and their projected evolution up to 2030.
- **Comparative Analysis of Different Technologies:** The paper's analysis of FCBs and BEBs in comparison ICEBs offers valuable insights into the trade-offs between energy consumption and GHG emissions. This comparative approach is crucial for policymakers and stakeholders in making informed decisions about sustainable urban transportation.
- **Future Projections and Scenario Analysis:** The projection to 2030 and the analysis of how these technologies might evolve over time adds forward-looking value to the research. It aids in understanding the long-term implications of adopting different bus technologies.

- Contribution to Global Decarbonization Efforts: Given the global urgency to reduce GHG emissions and transition to cleaner energy sources, the findings are relevant not just for Saudi Arabia but also for other regions with similar climatic and economic conditions.

- The finding that BEV busses perform worse, in terms of GHG emissions, than FCEV busses in the use phase (fuel cycle) is somewhat surprising, in many studies I have seen (often focussing on passenger cars and Europe though) the known pattern is that BEV perform better in the use phase and therefore break even after a few years of use. However, this finding depends on the assumptions made for the electricity mix and pathways for hydrogen production selected, which I fear are somewhat biased, see also concerns raised below.

Thank you for raising this important point. Your observation is correct that in many studies, particularly those focusing on passenger cars in Europe, BEVs are often found to perform better in terms of GHG emissions during the use phase. This is largely attributed to Europe's relatively cleaner electricity mix, which significantly impacts the environmental performance of BEVs.

However, our study presents a different context – Saudi Arabia, where the energy landscape is markedly different from that of Europe. In Saudi Arabia, the electricity mix is heavily reliant on fossil fuels, which influences the GHG emissions associated with BEVs. This distinction is crucial and underpins the unique findings of our research.

The apparent discrepancy in the performance of BEVs in our study compared to European-focused studies highlights the significant impact of local energy sources and production methods on the environmental impact of electric vehicles. This reinforces the need for region-specific LCAs to accurately determine the best decarbonization strategies. Our research underscores the importance of contextualizing LCA results to local conditions, as the optimal solutions for decarbonization can vary greatly depending on the geographic and energy context.

Therefore, the findings of our study are not only relevant to Saudi Arabia but also serve as a reminder of the broader significance of conducting LCAs in diverse energy environments. This approach is essential to understanding the global landscape of sustainable transportation solutions and emphasizes that there is no one-size-fits-all answer to decarbonization.

Our study could contribute to the growing body of knowledge by providing insights specific to the Saudi Arabian context, thereby enhancing the understanding of how local energy mixes can influence the environmental performance of different transportation technologies.

- It seems that the method as such is not novel, but extended to more detail for busses compared

to previous studies, albeit based on simple extrapolation of weight-based patterns from cars. It is not discussed to what extent this is realistic.

LCA (Life Cycle Assessment) is a well-established methodology with detailed and specific standards. This study strictly adheres to the guidelines set forth by ISO 14040 and 14044. While we do not innovate the LCA methodology itself, we bring novelty in its application to bus systems, particularly in the context of Saudi Arabia. This is a distinctive shift from traditional LCA applications which focus on cars. Our research introduces an adapted LCA methodology, specifically developed for bus systems, enhancing its relevance to our specific study area.

Furthermore, the uniqueness of our work is highlighted by our advanced computation and simulation techniques for gathering primary data, which are elaborated in the 'Methodology' section. We have gathered unique data that includes the hydrogen PEM Fuel Cell Bus (FCB) material composition, air conditioning heating load, shipping phase emissions and energy use, and refueling station infrastructure. These factors are pivotal to our study, providing fresh perspectives and insights in LCA for transportation systems.

Our approach extends beyond a mere weight-based extrapolation from car models. To elucidate this, we invite you to refer to the original supplementary material, specifically lines 28-57, and we move it to manuscript now (lines 123-146, page 6-7):

“While the primary distinction among different bus types lies in their powertrain and energy storage systems, both vehicle types share common body components such as doors, windows, seats, instrument panels, and controls, as well as similarities in chassis components, including brakes, suspension, wheels, and tires, and bumpers [34-36].

The construction of an FCBs predominantly relies on the component ratios of BEBs, with adjustments made for the powertrain and storage system. Given the Toyota Sora's total weight of 12,464 kg, it was assumed that the material distribution, excluding the powertrain, mirrors that of BEBs. For propulsion purposes, the Sora is equipped with four battery modules (each with a 6.5Ah capacity), rather than for emergency power supply. Prismatic NiMH battery modules from Panasonic, each comprising six 1.2V cells in series, are utilised by Toyota [37]. The module features a rated voltage of 7.2V, a 6.5Ah capacity, a weight of 1.04 kg, and dimensions of 19.6 mm (W) x 106 mm (H) x 275 mm (L) [37]. This yields an estimated total battery weight of 214 kg for the Toyota Sora. Concerning the PEM FC stack, the Toyota Mirai's PEM FC stack (114 kW) weighed 56 kg [37]. Considering the Sora was fitted with two PEM FCs, it was inferred that the Sora's PEM FC stack weighs 112 kg [38]. Regarding high-pressure (HP) hydrogen tanks, the Sora is equipped with 10 tanks rated at 70 MPa. Given a reported combined weight of approximately 87.5 kg for two HP hydrogen tanks, the estimated weight for ten hydrogen tanks was 437.5 kg [39]. Furthermore, following the GREET's life-cycle coefficient of the fuel cell (FC) stack and auxiliary system data which offers a weight-to-power ratio of 1.27kg/kW, the weight of the FC BOP can be projected to be 179 kg [36].”

Is the work convincing, and if not, what further evidence would be required to strengthen the conclusions?

I am missing detail on the assumptions made for being able to fully judge the validity of conclusions. Main items listed in the concerns below.

On a more subjective note, do you feel that the paper will influence thinking in the field?

Personally I am not sure it will. The fact that the choice of fuel/energy pathways is decisive for the life cycle balance of different vehicle fuel and powertrain options is well known. If the authors could substantiate better the choice of their pathways and prove the case that for H₂, better production pathways are and remain available than for electricity, this could be of interest. However, it is mentioned several times in the paper that the Saudi-Arabian context is rich in sun, therefore a carbon-intensive production of electricity seems to be a choice, not a given.

Thank you for your insightful observation regarding the impact of fuel and energy pathway choices on the life cycle balance of different vehicle options.

Our research particularly focuses on the energy landscape in Saudi Arabia, which is currently undergoing a significant transition. While the region shows a strong potential for solar energy, the development of solar power plants is still in progress. As part of Saudi Arabia's Vision 2030, there is a strategic plan to generate at least 50% of the country's energy from renewable sources, predominantly solar power. This ambitious goal reflects a shift towards sustainable energy practices.

However, it is important to note that, as of now, Saudi Arabia's energy sector remains substantially reliant on fossil fuels. This reality forms the basis of our study and is critical to understanding the context in which we have evaluated the life cycle balance of different vehicle options. This reliance on fossil fuels and its implications for our study are explicitly mentioned in our manuscript (section 4.6.4, page 22-23). For a detailed discussion on this, we refer you to reference 53-55, specifically in manuscript (lines 622-631, page 22-23), where we delve into the current state of Saudi Arabia's energy sector and its influence on our findings.

“Utilising a multitude of data sources [48-50], this study conducted an evaluation of Saudi Arabia's electricity generation landscape for 2022, predicated on the assumption that it remains consistent with the 2021 data. The energy portfolio is primarily composed of NG, producing 215.93 TWh, followed by oil at 139.86 TWh, and solar energy contributing a modest 0.83 TWh [48-50]. In accordance with the Saudi Green Initiative's 2030 vision, the target is to increase the share of renewable energy to 50% by 2030 [51].”

Given this context, in the short term, the production of blue and grey hydrogen from natural gas appears to be the most viable option in Saudi Arabia.

Please feel free to raise any further questions and concerns about the paper.

- 1. Main assumptions on operation conditions and bus specifications are not specified. This regards, among others, the drive profiles and mileage assumptions used, bus size and motorization, battery and fuel tank capacities etc., as well an explanation under what conditions calculations apply (e.g., real world operation, or some cycles).

- 2. I just saw one hint in figure 1 that the busses run two trips of 70km per day and are used for 4 hours a day. This assumption seems to be very low. Given the investment cost of busses I would expect an economically viable activity of at least double the trips per day.
- 3. If busses are to do 70km trips with a one hour stop at a given depot in-between, battery capacity (as well as H2 tank size) can be quite low. Intuitively, I would expect the BEV battery to cause less production burden in this case.
- 4. Also the time frame of vehicle use considered is not specified. I would have expected that BEV fare better, in particular over long use periods and large lifetime activity, given their superior energy efficiency. Only kwh/km are specified in the paper so it is hard to check.
- 5. The fact that urban busses have specific operation profiles is not discussed. For example, what is assumed with regard to refuelling/recharging/infrastructure? It seems that the build up of dedicated infrastructure is included in the calculations, but is there any reflection of the fact that urban busses typically have standardized routes and can be charged in a dedicated depot, thus not necessarily requiring the build-up of wide-spread infrastructure?

These 5 comments are all related to bus operation conditions and specifications, and we would like to reply to them together here.

- First of all, in our initial manuscript, detailed specifications of the buses, including size, motorization, battery, and fuel tank capacities, are comprehensively listed under original Supplementary Information I.1, Table 1. For enhanced accessibility, this information is now also included in the main manuscript (lines 497 – 500, page 19). Here, we provide a summary of the specifications for the three bus types examined:

“Table 2 lists the specifications for the three types of buses. The FCBs considered in this research were modelled after the Toyota Sora, while the BEBs were referenced to the BYD K9 [14]. For the ICEBs, the Volvo B8RLE served as the basis.”

- Secondly, initially presented in Fig. 1 (right bottom corner), we have expanded on the bus operation conditions in the manuscript (lines 109-119, page 5):

“In an urban bus transportation framework, the study focuses on a fleet comprising 20 buses—18 for regular operation and 2 as reserves. Each bus is expected to operate for 10 years, traveling a total of 508,080 kilometers. The study's scope encompasses a specific urban trajectory from Shimeisy Police Station to Al Haram in Makkah, with empirical data obtained through collaboration with the bus company. The fleet adheres to a structured schedule, where each bus completes two rounds daily. Operational design involves a 2-hour active period for each round, interspersed with a 3-hour rest period, thereby optimizing fleet performance. Service frequency is set at 20-minute intervals, spanning from 08:00 to 22:00, to cater to the year-round Umrah-related demand peaks.”

- Our study's bus operation model is grounded in a real-case scenario from the Mecca region, in collaboration with the Saudi Public Transportation Company. Although bus economics is beyond the scope of this paper, we have made certain assumptions detailed in the Supplementary Information (lines 186-205, page 8). These include energy consumption rates for different bus types and considerations regarding their operation in Makkah's climate.

“Drawing from established literature, FCBs have been reported to have an energy consumption of 5.5 kg H₂/100km, excluding AC and auxiliary systems [32, 33]. The operating temperature range of the PEM FC stack falls between 60°C and 80°C, considerably higher than the average outdoor temperature in Makkah [34-37]. Thus, it was postulated that the PEM FC stack’s operation would remain unaffected by outdoor temperature, implying that the baseline energy consumption would similarly remain consistent regardless of external temperature.

In comparison, BEBs have been identified to consume 0.9 kWh/km under the same conditions [32, 38, 39]. Extensive studies suggest that a temperature range of 25°C to 40°C is deemed optimal for charging and discharging lithium-ion batteries [40-45]. Consequently, it was postulated that an ambient temperature of 39°C would not influence the baseline performance of BEBs.

For buses powered by ICE, energy consumption without heating, ventilation, and AC system has been estimated to be 30 l_{diesel}/100 km [46-50]. Utilising the pump to well energy conversion efficiencies for FCBs, BEBs, and ICEBs are shown in Table. S4, alongside the quantified cooling load pertaining to the AC system and the bus’s baseline energy consumption, determinations can be drawn for the final requisite energy for bus operation. The life-cycle coefficient of fuels has been procured from the GREET model.”

- The assumed daily run of 139.2 km for each bus, culminating in 508,080 km over a 10-year lifespan, aligns with the average life mileage of buses documented in references (Ellingsen et al., 2022; Nordelöf, Romare, & Tivander, 2019)), and corroborated by practical information.

- Why only 139.2 km/day?

We agree with the Review's point that buses have standardized routes and do not need a wide-spread infrastructure. We have assumed the minimum amount of infrastructure build-up enough to meet the bus fleet's demand, which is 1 charging station for BEB and 1 hydrogen station for FCB. Details of the infrastructure are described in previous Supplementary III.4 and IV.3. Currently, the details of this infrastructure are moved to the manuscript (lines 612-630 page 22, and 681-692, page 25), covering aspects like the Hydrogen Refuelling Station (RFS) and Super-fast Charging Stations, their capacity, and the material breakdown of infrastructure components.

1. Lines 601-620, page 22: “4.6.3 Hydrogen refuelling station infrastructure

The materials list for the RFS inventory in this study was based on a bill of materials prepared by Mailänder in 2003 for a HydroStatoil model hydrogen RFS in Reykjavik [68].

The hydrogen PEM FCBs under study have a 600-litre hydrogen tank (equivalent to 23.46 kg of hydrogen) and consumes an estimated 9.7 kg per 100 km with the AC system turned on. The capacity of the designed hydrogen RFS was determined based on the total distance travelled by 18 buses per year, the frequency of hydrogen refuelling, and the fuel consumption per 100 km. The Makkah hydrogen RFS features a daily capacity of 259 kg, adeptly accommodating the 245 kg/day hydrogen refuelling requirements of the buses. The necessary material breakdown list of the infrastructure components such as building foundations and compressors, as outlined in Table S6 in Supplementary Note 3.

Table S7 in Supplementary Note 3 provides crucial data on the expected service life of the essential components of the RFS infrastructure, including RFS, compressors, storage tanks, trailers, and hydrogen pipelines. The study assumed a 10-year lifetime for the entire bus fleet. All the components mentioned above are expected to have a service life of at least 10 years. Therefore, it can be inferred that the missing components of the fuelling station infrastructure need not be replaced within the specified 10-year period.”

2. Lines 672-6833, page 25: “4.6.6 Super-fast charging stations

The charging station model used in this study is based on the work of Zhao et al. (2021) and was adapted to maintain the systematic operation of the bus fleet [74]. The charging station comprises six chargers, three power supply units, and a control unit, enabling up to six buses to be charged concurrently with a maximum output power of 350 kW per charger. However, to ensure safety, the BEBs in this study have a charging power of 150 kW and require a full charging duration of 2-2.5 hours. Each power unit can serve up to two chargers. The control unit includes a communication unit that acts as a centralised system for power and load management of the charging station. The composition of materials is detailed in Supplementary Note 4, Table S8, with emission factors for material production sourced from the GREET database.”

- I seems that, for hydrogen, two rather advanced production routed (including also CCS, which to my information is not yet a widely applied standard technology) are used. In contrast, for electricity, the paper uses grid mix. This seems a somewhat biased assumption. Either electricity grid mix needs to be compared to average today H2 production pathway, or for both ambitious assumptions should be made for a fair comparison (e.g., marginal electricity production for BEV, potentially taking into account the option of providing dedicated renewable energy plants for producing electricity close to depot).

Thank you for highlighting the concern regarding our fuel production assumptions. Our assumptions of the fuel production route are based on real-case scenarios in Saudi Arabia, with data and comments provided by our industrial partners. In 2022, there are very limited renewable plants running in the country and EVs will have to use grid power in practice. This is evidenced by the data presented in our manuscript (lines 624-626, page 22):

“The energy portfolio is primarily composed of NG, producing 215.93 TWh, followed by oil at 139.86 TWh, and solar energy contributing a modest 0.83 TWh [48-50].”

This translates to only about 0.23% of the electricity coming from renewable sources. Additionally, there are no significant renewable energy plants in the vicinity of Makkah in 2023, reinforcing our rationale for the current scenario.

On the other hand, grey and blue hydrogen production is already at a commercial scale in Saudi Arabia and is possible to fuel a small bus fleet. Furthermore, we did take the expectation to achieve 27.3GW of renewable power capacity into consideration for the 2030 scenario, which shows in our assumptions that renewable energy will account for 50% of the national power supply, which renders significantly BEV's lifecycle emissions decrease in the 2030 scenario already. There is no contradiction to Review's point of view.

- The energy aspect is not discussed appropriately. Figure 5 shows that the lease GHG intense option (blue H2 FCEV) consumes about 20% more energy than grey H2 FCEV or BEV in 2022, and the gap versus BEV grows to almost 40% by 2030. This could raise some concerns over energy availability and affordability of the preferred solution. Moreover, in the conclusions, efuels are mentioned (without any evidence or analysis) as a promising option to decarbonize ICE, whereas it is known from several studies that efuel and ICE are both very energy inefficient options. Energy efficiency should be better discussed in the paper.

The reviewer's comment and observation are highly appreciated and we acknowledged the importance of energy availability and affordability for future development. However, this paper's main focus is on the potential of decarbonization, therefore life cycle emission reduction is our main discussion point. Nevertheless, discussions about each buses' energy consumption can be found in our original manuscript on pages 8-11. As the reviewer pointed out, Fig 6 also displays the energy consumption data clearly which readers can draw information from.

To address this, we will expand our discussion in the revised manuscript to more explicitly analyze the implications of these differences in energy consumption, particularly in relation to energy availability and the feasibility of scaling up these technologies.

Regarding the mention of e-fuels in the conclusions, we would like to clarify that this was intended to highlight potential options for reducing lifecycle emissions in internal combustion engines (ICE). It was not our intention to imply that e-fuels are an energy-efficient solution. We recognize that the energy efficiency of e-fuels and ICE is a complex and ongoing area of research. Therefore, in our revised manuscript (lines 458-467, page 18), we will clarify that the mention of e-fuels is speculative and based on their potential for emission reduction rather than energy efficiency. We aim to ensure that this distinction is clear to avoid any misunderstanding about the scope of our study and the current state of research in this area.

lines 458-467, page 17:

“Under all circumstances, ICEBs bear the highest life-cycle carbon footprints, a consequence of the immense emissions generated during bus operation. Therefore, it becomes increasingly clear that curbing GHG emissions necessitates an essential focus on reducing the carbon footprint of fuels, highlighting the need for low-carbon fuels, like "e-fuels". Nevertheless, the nascent stage of these fuels necessitates further LCA studies to deepen our understanding of the decarbonization potential inherent in the overall system. Furthermore, an evaluation of the energy efficiency of these alternatives is indispensable, ensuring a holistic approach to their integration into the energy landscape.”

- In Table 1, for 2022, the share of recycled steel and aluminium is different for the three bus types (while in 2030 it is the same). Why is this the case? I expect these to be the materials used for the vehicle body, in that case shouldn't it be similar?

In 2022 scenario, the three types of buses are assumed to be made in different countries and imported to Saudi Arabia. Therefore, the material recycling rates are set specific to the original countries. In 2030 scenario, we have assumed that all buses to be made in Saudi Arabia; hence the material recycling rates are the same for all bus types. Details are described in original SI section I.2 Recycled materials used in buses. We have updated our manuscript to include these details and clarifications. The relevant information is now more prominently presented in the manuscript (lines 731-739, page 26) instead of the supplementary information, ensuring that readers can easily access and understand our methodology and assumptions.

Lines 731-740: "Considering the majority of buses in operation within Saudi Arabia were imported, the study posited that ICEBs originate from Sweden, BEBs from China, and FCBs from Japan. As a result, production-related data were assumed to be country-specific. The estimated shares of recycled steel in vehicles for Japan, China, and Sweden were respectively 23.2%, 10.1% and 41.6% [60]. In addition, the shares of recycled aluminium in these countries were assumed to be 99.4%, 19.2%, and 70.9%, respectively [61]. In the 2030 scenario, all three types of buses, manufactured in Saudi Arabia, will utilize data from the GREET model."

- Figure axes labels come out very small (particularly figure 3).

We have enlarged the axes labels in Fig. 3.

We would also be grateful if you could comment on the appropriateness and validity of any statistical analysis, as well the ability of a researcher to reproduce the work, given the level of detail provided.

Main assumptions are not specified, see previous concerns. To my view, the analysis is not reproducible with the data provided.

We have clearly outlined all assumptions in our study in the main text and provided extensive details in the Supplementary Information (SI). Our methodology and data are comprehensively described in the manuscript, with supplementary details in the SI, ensuring transparency and replicability. While all relevant citations are listed in the SI, we welcome specific inquiries for further clarification or detail. For example, the SI includes a detailed analysis of fuel pathway scenarios and the methodology for calculating air conditioning cooling loads.

We are confident that our work results are reproducible using our model set, given the same set of data. As we mentioned previously, the detailed methodology can be found in SI for reproduction purposes. We are also confident that the model is applicable to other similar LCA analyses.

References:

- Aramco. (2019). Saudi Aramco and Air Products inaugurate Saudi Arabia's first hydrogen fueling station. Retrieved from <https://www.aramco.com/en/news-media/news/2019/hydrogen-inauguration>
- Ellingsen, L. A.-W., Thorne, R. J., Wind, J., Figenbaum, E., Romare, M., & Nordelöf, A. (2022). Life cycle assessment of battery electric buses. *Transportation Research Part D: Transport and Environment*, 112, 103498.
- Hasan, S., & Shabaneh, R. (2021). The Economics and Resource Potential of Hydrogen Production in Saudi Arabia. *KAPSARC: Riyadh, Saudi Arabia*.
- Iannuzzi, L., Hilbert, J. A., & Lora, E. E. S. (2021). Life Cycle Assessment (LCA) for use on renewable sourced hydrogen fuel cell buses vs diesel engines buses in the city of Rosario, Argentina. *International Journal of Hydrogen Energy*, 46(57), 29694-29705.
- Lee, D.-Y., Elgowainy, A., Kotz, A., Vijayagopal, R., & Marcinkoski, J. (2018). Life-cycle implications of hydrogen fuel cell electric vehicle technology for medium-and heavy-duty trucks. *Journal of Power Sources*, 393, 217-229.
- Lucas, A., Neto, R. C., & Silva, C. A. (2012). Impact of energy supply infrastructure in life cycle analysis of hydrogen and electric systems applied to the Portuguese transportation sector. *International Journal of Hydrogen Energy*, 37(15), 10973-10985.
- Nordelöf, A., Romare, M., & Tivander, J. (2019). Life cycle assessment of city buses powered by electricity, hydrogenated vegetable oil or diesel. *Transportation Research Part D: Transport and Environment*, 75, 211-222.
- Velandia Vargas, J. E., Falco, D. G., da Silva Walter, A. C., Cavaliero, C. K. N., & Seabra, J. E. A. (2019). Life cycle assessment of electric vehicles and buses in Brazil: effects of local manufacturing, mass reduction, and energy consumption evolution. *The International Journal of Life Cycle Assessment*, 24, 1878-1897.

Reviewers' comments:

Reviewer #1 (Remarks to the Author):

Dear Authors and Editor,

After reviewing the revisions made to the manuscript and reevaluating the updated submission, I am pleased to accept the changes and agree that the manuscript can be published in its current form. The modifications effectively addressed my previous comments, enhancing the manuscript's coherence, clarity, and depth of analysis. I appreciate the alignment of the title, abstract, and introduction with the main focus on decarbonising urban bus transportation in Saudi Arabia, which provides clarity on the subject matter. I concur with its publication in the present form, considering all the enhancements and significant improvements made to the manuscript. This work makes an important contribution to the field, and I look forward to its publication.

Thank you to the authors for considering my suggestions and for their commitment throughout the review process. Congratulations on your excellent work, and best wishes for your future research.

Sincerely,
Reviewer

Reviewer #3 (Remarks to the Author):

Thanks for inviting me to review the revised version of the paper.

The authors have made substantial effort to revise the paper, and have added substantiation and clarifications in many respects. Many of my comments have been addressed satisfactorily. In particular, shifting parts of the annexes to the main paper and providing additional explanation has helped making the paper substantially more understandable stand-alone.

As a minor comment, it would help the comparison of the present results with other studies if main parameters, for example the tank-to-wheel vehicle efficiencies (Kwh of energy per vkm) and of Well-to-tank parameters (MJ of energy expended and kgCO₂ emitted per km, or even better, per MJ of fuel and energy produced) were also given in a table. I attempted a comparison for 2022 from the data given in Fig. 6, and orders of magnitude seem plausible. But having access to the numbers, even in an annex, would make this substantially easier.

Unfortunately, the conceptual problem with the fuel-propulsion type combinations chosen for comparison has not been resolved, and I still feel the comparison and thus the conclusions are biased. In particular, comparing fuel cell busses fuelled exclusively with blue hydrogen, which to my understanding is a niche product in the present Saudi Arabian energy mix, against battery electric busses using grid mix electricity, is not a fair comparison. The authors argue that "grey and blue hydrogen production is already at a commercial scale in Saudi Arabia and is possible to fuel a small bus fleet". But I assume that, likewise, it would be possible to run a small BEV bus fleet exclusively on renewable electricity, and that this would be the correct comparison to a blue hydrogen fleet, ideally in addition to grid mix electricity BEV busses, to span the full range.

Alternatively, it would be an equilibrated comparison for 2022 to show grid-electricity-mix BEV versus average Saudi Arabian H₂ mix FCEV busses. Similarly, for 2030 the comparison should be projected electricity mix in 2030 (as done) compared to projected average H₂ available in Saudi Arabia in 2030 (which is not done, instead exclusively blue or grey H₂ are used).

As is, the authors credit the GHG reduction from blue hydrogen to transport – but even if it is possible

to use the small existing amount of blue H₂ for fuelling busses, this would mean crowding out other demand for blue H₂, and, consequently, shifting the emissions elsewhere, as long as no large-scale market for hydrogen exists.

Last not least, the economics of the different options, as well as other boundary conditions (e.g. how/where to sequester carbon etc) remain to be investigated before identifying a preferred solution for decarbonisation of bus transportation. This should shortly discussed and acknowledged in the paper.

Ms. Ref. No.: COMMSENG-23-0358B

Title: Solutions for Decarbonizing Urban Bus Transport in non-OECD Countries: A Life Cycle Case Study in Saudi Arabia

Detailed Response to Reviewers

The authors express their gratitude to the editors and reviewers for dedicating their valuable time to review our manuscript titled “Solutions for Decarbonizing Urban Bus Transport in non-OECD Countries: A Life Cycle Case Study in Saudi Arabia”. We sincerely appreciate the insightful comments and suggestions provided by the reviewers and the editor, which have significantly enhanced the quality of our paper. In response to this constructive feedback, we have made further revisions and improvements to the manuscript. Specifically, we have organized the results data and added new scenarios to enrich the manuscript, namely, BEBs powered entirely by renewable energy and FCBs using a mix of hydrogen. We believe these modifications have substantially solidified the foundation of our manuscript and effectively addressed the concerns raised by the reviewers.

Here are our responses to the reviewers' comments:

Reviewers' comments:

Reviewer #1 (Remarks to the Author):

Dear Authors and Editor,

After reviewing the revisions made to the manuscript and reevaluating the updated submission, I am pleased to accept the changes and agree that the manuscript can be published in its current form. The modifications effectively addressed my previous comments, enhancing the manuscript's coherence, clarity, and depth of analysis. I appreciate the alignment of the title, abstract, and introduction with the main focus on decarbonising urban bus transportation in Saudi Arabia, which provides clarity on the subject matter. I concur with its publication in the present form, considering all the enhancements and significant improvements made to the manuscript. This work makes an important contribution to the field, and I look forward to its publication.

Thank you to the authors for considering my suggestions and for their commitment throughout the review process. Congratulations on your excellent work, and best wishes for your future research.

Sincerely,

Reviewer

The authors are grateful for the encouraging comments from the reviewer. There is no question that needs to be addressed.

Reviewer #3 (Remarks to the Author):

Thanks for inviting me to review the revised version of the paper.

The authors have made substantial effort to revise the paper, and have added substantiation and clarifications in many respects. Many of my comments have been addressed satisfactorily. In particular, shifting parts of the annexes to the main paper and providing additional explanation has helped making the paper substantially more understandable stand-alone.

As a minor comment, it would help the comparison of the present results with other studies if main parameters, for example the tank-to-wheel vehicle efficiencies (Kwh of energy per vkm) and of Well-to-tank parameters (MJ of energy expended and kgCO₂ emitted per km, or even better, per MJ of fuel and energy produced) were also given in a table. I attempted a comparison for 2022 from the data given in Fig. 6, and orders of magnitude seem plausible. But having access to the numbers, even in an annex, would make this substantially easier.

Thank you very much for your inquiry regarding the vehicle efficiencies and the detailed analysis of the life-cycle stages presented in our study.

For the tank-to-wheel vehicle efficiencies, we have provided an explanation in the Supplementary Information, specifically in Note 2, Table S5. Additionally, the tank-to-well vehicle efficiencies are illustrated in Figure 4 on page 10 of our manuscript, offering a clear depiction of these values.

Regarding the life-cycle stages, including the energy expended and kgCO₂-eq. emissions per kilometer, these results are presented in tabular form in the Supplementary Information. Specifically, you can find this detailed information in Supplementary Note 10, Table S14, and Table S15, on pages 19-20. These tables offer an in-depth view of the well-to-wheel and vehicle results for all scenarios examined in our study, providing a comprehensive understanding of the environmental impacts associated with each scenario.

Unfortunately, the conceptual problem with the fuel-propulsion type combinations chosen for comparison has not been resolved, and I still feel the comparison and thus the conclusions are biased. In particular, comparing fuel cell busses fuelled exclusively with blue hydrogen, which to my understanding is a niche product in the present Saudi Arabian energy mix, against battery electric busses using grid mix electricity, is not a fair comparison. The authors argue that “grey and blue hydrogen production is already at a commercial scale in Saudi Arabia and is possible to fuel a small bus fleet”. But I assume that, likewise, it would be possible to run a small BEV bus

fleet exclusively on renewable electricity, and that this would be the correct comparison to a blue hydrogen fleet, ideally in addition to grid mix electricity BEV busses, to span the full range.

We are grateful for your insightful feedback and further opportunity to clarify the comparison framework within our study. The choice to evaluate FCBs powered by blue hydrogen against BEBs powered by the grid mix electricity in the Saudi Arabian context was carefully made, reflecting not just the current energy landscape but also anticipated future developments. The establishment of the first hydrogen refueling station in 2019 to support a fleet of six Mirai passenger cars underscores the readiness and ambition of Saudi Arabia to integrate hydrogen, particularly blue and grey hydrogen, into its transportation sector by 2022 (Aramco, 2019).

Acknowledging the limitations in renewable energy infrastructure within the Makkah region as of 2022, we introduced a theoretical scenario to explore the potential of BEBs powered entirely by renewable energy, specifically solar energy, for 2022 and 2030. This scenario is detailed in our manuscript (lines 302-312, pages 11-12) and is visually represented in Figure 6, which outlines the life-cycle energy consumption and GHG emissions for hypothetical bus fleets.

lines 302-312, pages 11-12: “Currently, the Makkah region faces a significant deficit in renewable energy infrastructure, presenting challenges for the deployment of BEBs that depend solely on renewable energy. This study investigates not only a feasible bus fleet model but also explores a speculative scenario in which BEBs are powered entirely by renewable energy sources (100% green BEBs) for the years 2022 and 2030, aiming to ascertain their ultimate carbon mitigation efficiency relative to FCBs utilizing blue hydrogen. Figure 6 provides a comprehensive portrayal of life-cycle energy consumption and GHG emissions for anticipated bus fleets in 2022 and 2030, alongside the total life-cycle effects of a conjectural fleet of BEBs powered purely by solar energy, discounting contributions from photovoltaic (PV) infrastructure.”

Notably, our analysis suggests a significant reduction in energy usage and GHG emissions for BEBs powered solely by solar energy compared to FCBs utilizing blue hydrogen, highlighting the potential for substantial decarbonization within the sector. Discussion on the 2022 scenario for 100% green BEBs is detailed in our manuscript (lines 337-342, page 13).

lines 338-343, pages 13: “In a hypothetical scenario where BEBs are powered entirely by solar energy, the forecasted energy consumption and CO₂-eq emissions are expected to decrease to 9.82 MJ km⁻¹ and 0.26 kg CO₂-eq km⁻¹, correspondingly. This denotes a 54% reduction in energy usage compared to grey hydrogen FCBs and a 69% decrease in GHG emissions relative to blue hydrogen FCBs.”

Discussion and conclusions regarding the 2030 scenario for 100% green BEBs are detailed in our manuscript, specifically lines 408-411 on page 15 and lines 487-495 on page 18.

lines 408-411, pages 15: “Remarkably, a hypothetical 100% green BEBs scenario emerges as the most GHG efficient model in 2030, achieving GHG emissions that are 79% lower than grid BEBs and 72% lower than purely blue hydrogen FCBs and surpassing the grid BEB by 40% in energy efficiency.”

Conclusion (lines 487-495, page 18): “In a theoretical framework, BEBs powered entirely by renewable energy emerge as the premier option for carbon reduction and energy efficiency. Nevertheless, the practicality of exclusively using renewable energy for BEBs depends on factors such as the availability of renewable resources and Saudi Arabia's strategic progress in the

renewable energy sector, especially the allocation of renewable energy to the transport sector. While BEBs powered by renewable sources exhibit zero emissions in the fuel cycle, the wider environmental and energy impacts associated with the infrastructure for renewable energy generation, including PV systems, necessitate thorough investigation.”

In our manuscript, the methodology section was expanded to include lines **844-854**, page 30.

“4.9 Renewable Energy-Powered BEBs: Scenarios for 2022 and 2030

This study evaluates the carbon reduction potential of BEBs powered solely by renewable energy, juxtaposed with a scenario of 100% blue hydrogen, for the years 2022 and 2030. It assumes that all electricity from the “tank-to-wheel” phase is sourced from solar energy. However, the environmental impact of PV plant infrastructure is not considered within this analysis. For both scenarios, the electricity consumed in the vehicle manufacturing phase and RFS infrastructure construction is drawn from the grid.”

Alternatively, it would be an equilibrated comparison for 2022 to show grid-electricity-mix BEV versus average Saudi Arabian H2 mix FCEV busses. Similarly, for 2030 the comparison should be projected electricity mix in 2030 (as done) compared to projected average H2 available in Saudi Arabia in 2030 (which is not done, instead exclusively blue or grey H2 are used).

To complement this comparison and ensure fairness, we also examined a scenario involving FCBs powered by a mix of hydrogen sources by 2030, reflective of projections that foresee a dominant share of blue over grey hydrogen. According to forecasts by SWP, it is expected that the ratio of blue hydrogen to grey hydrogen will be 98% to 2% by the year 2030 (Ansari, 2022). Analysis of the Mixed Hydrogen Scenario is detailed in our manuscript (lines 412-415, page 15), with conclusions presented in lines 472-473, page 17.

lines **412-415**, page 15: “For 2030, FCBs powered by a blend of 98% blue and 2% grey hydrogen are projected to produce CO₂-eq emissions of 0.57 kg km⁻¹ and require 24.74 MJ km⁻¹ of energy, paralleling the environmental impact of FCBs powered solely by blue hydrogen.”

Conclusion (lines **472-473**, page 17): “The performance of 2030 mixed hydrogen scenario is nearly identical to pure blue hydrogen scenarios.”

The methodological details for the mixed hydrogen scenario are elaborated in the manuscript, lines **855-859**, on page 31.

“4.10 2030 mixed hydrogen scenario

For a balanced assessment against grid-connected BEBs in 2030, this study delves into a scenario envisioning FCBs powered by a hybrid hydrogen mix. According to projections, this mix is anticipated to comprise 98% blue and 2% grey hydrogen [66].”

As is, the authors credit the GHG reduction from blue hydrogen to transport – but even if it is possible to use the small existing amount of blue H2 for fuelling busses, this would mean crowding out other demand for blue H2, and, consequently, shifting the emissions elsewhere, as long as no large-scale market for hydrogen exists.

Thank you for your insightful comments and for highlighting the importance of considering the potential crowding-out effect of blue hydrogen demand. We acknowledge the concerns regarding the crowding-out effect and its implications for the hydrogen market. However, our analysis takes into account the strategic expansion plans for hydrogen production in Saudi Arabia. The Kingdom's significant investments in this sector are not solely focused on meeting domestic demands but are also aimed at securing a pivotal role in the global hydrogen economy (Hassan et al., 2023). This ambition is supported by substantial efforts to increase production capacities, which we believe can effectively mitigate the concerns about the crowding-out effect.

Furthermore, our study implicitly considers the dynamic and evolving nature of energy markets and the potential for technological and infrastructural advancements. By focusing on blue and grey hydrogen, we aim to highlight immediate opportunities and challenges within the Saudi context, acknowledging that the energy landscape is subject to rapid changes influenced by policy, innovation, and market demands.

Similarly, the purpose of our research on grid-connected BEBs extends to ensuring that the reduction of GHG emissions achieved through renewable energy sources is not solely attributed to the transport sector. Additionally, our study explores scenarios involving 100% blue hydrogen and grey hydrogen intending to establish a benchmark for the industry, defining a limit value that not only guides industry stakeholders towards best practices but also provides crucial information for policymakers to develop informed and impactful strategies.

In summary, our approach to comparing different fuel-propulsion types is grounded in a nuanced understanding of the current and future Saudi Arabian energy landscape, incorporating both existing limitations and speculative scenarios to provide a balanced and comprehensive analysis of sustainable transportation solutions.

In addition, the **abstract** and **conclusion** of our **manuscript** have been modified accordingly.

Abstract: "With the vast majority of the global population, non-OECD countries such as Saudi Arabia, are facing a harder battle against carbon emissions. Saudi Arabia's urban bus transportation is under pressure to decarbonize, and new energy vehicles can mitigate emissions. This life-cycle assessment illustrates that, fuel cell buses (FCBs) using blue hydrogen can reduce emissions by 53.5% compared to conventional buses, despite a 19.7% increase in energy use from carbon capture and storage (CCS) system. The battery electric buses (BEBs), affected by Saudi's energy mix and lithium-ion batteries manufacturing, only cut emissions by 16.7%. These climate benefits pivot significantly on energy sources, and the efficiencies of CCS and hydrogen production. With projected technological advancements and 50% renewable energy by 2030, FCBs and BEBs could achieve further carbon reduction by 19.3%-51%, potentially lowering emissions to 0.16 kgCO₂-eq/km for BEBs powered entirely by renewables. However, FCBs consistently demonstrate lower environmental burdens, while BEBs manifest the highest impacts."

Last not least, the economics of the different options, as well as other boundary conditions (e.g. how/where to sequester carbon etc) remain to be investigated before identifying a preferred solution for decarbonisation of bus transportation. This should shortly discussed and acknowledged in the paper.

We thank the reviewer for highlighting the importance of exploring the economic aspects and other boundary conditions such as carbon sequestration options, which are indeed critical for evaluating the sustainable pathways for bus transportation. To address this, we included a

paragraph in conclusion that briefly outlines the economic implications and boundary conditions necessary for the implementation of these technologies. This acknowledgment is clearly stated in our revised manuscript (lines 520-524, pages 18-19).

lines 520-524, pages 18-19: “Furthermore, achieving sustainable bus transport solutions necessitates a multifaceted strategy, extending beyond environmental impacts to include economic assessments of bus alternatives and examination of boundary conditions like the efficacy of CCS in the blue hydrogen cycle, carbon sequestration capabilities, and the development of renewable energy infrastructure.”

We believe these adjustments and clarifications will address your concerns regarding the fairness and bias of our comparison. Our goal is to provide a nuanced and comprehensive evaluation of the potential roles of different bus technologies in a future sustainable transportation system, considering both current realities and future possibilities.

Once again, we appreciate your constructive feedback and the opportunity to enhance the robustness and relevance of our study.

References

- Ansari, D. (2022). The hydrogen ambitions of the Gulf States. *Stiftung Wissenschaft und Politik*.
- Aramco. (2019). Saudi Aramco and Air Products inaugurate Saudi Arabia’s first hydrogen fueling station. Retrieved from <https://www.aramco.com/en/news-media/news/2019/hydrogen-inauguration>
- Hassan, Q., Sameen, A. Z., Salman, H. M., Jaszczur, M., Al-Hitmi, M., & Alghoul, M. (2023). Energy futures and green hydrogen production: Is Saudi Arabia trend? *Results in Engineering*, 18, 101165.

REVIEWERS' COMMENTS:

Reviewer #3 (Remarks to the Author):

Thanks for inviting me to review the second revised version of the paper. Again, the authors have gone to great lengths to revise the paper. The request for additional data has been answered to my satisfaction. Moreover, scenarios for 100% renewable electricity-powered BEB (2022 and 2030) and for the hydrogen mix in 2030 have been added. This provides the basis for an unbiased comparison of all options. The additional outcomes show that, in terms of technology, BEB busses run on RES electricity are the most energy efficient solution and achieve the highest life-cycle GHG emission reductions, which confirms previous research results.

I have following further suggestions for minor edits:

- With the addition of the 100%RES scenario, it becomes implicitly clear that the conclusion of blue H2 busses as the most effective option for GHG emission reduction rests on the assumption that blue H2 will become abundantly available in Saudi Arabia, whereas RES electricity will not (at least by 2030). It should, therefore, be made clear why renewable electricity is not viable or feasible for Saudi Arabia. It seems that the barrier is not of technical nature, but lies with the supposed present energy policy strategy of Saudi Arabia to ramp up hydrogen but not renewable electricity. Unless I am mistaken (I screened the paper again in this regard, but no full read), it is not made sufficiently clear that this strategy exists, if and why Saudi Arabia is not planning to exploit renewable electricity, and how credible and persistent this strategy is out to 2030. A short description of the Saudi Arabian energy strategy, including suitable evidence, should be added (either to the introduction, or in section 2.3 on the fuel cycle) to show of what nature is the barrier for not being able to go for more renewable electricity consumption in transport, and why this is therefore considered "theoretical", "hypothetical" or not "viable/feasible" in the paper.

- In the Conclusions, authors use the terms "feasible bus fleets", or "viable bus fleets". I think this formulation is misleading, as on the fleet side, fully electric bus fleets are included in the viable solutions in the paper. It seems therefore that the authors refer to feasible or viable fuel/electricity production pathways, from which they exclude fully renewable electricity. Here, wording should be adapted to make clearer for the reader that it is not the fleet authors deem not viable (e.g., saying "feasible electricity supply").

- Another minor suggested edit regards the abstract, where new text has been added that reads: "With projected technological advancements and 50% renewable energy by 2030, FCBs and BEBs could achieve further carbon reduction by 19.3%-51%, potentially lowering emissions to 0.16 kgCO₂-eq/km for BEBs powered entirely by renewables." The formulation is unclear,

- Does it mean "... by 19.3% and 51% respectively" (i.e., 19.3% for FCB and 51% for BEV)?

- 0.16 kgCO₂/km is the only absolute emission mentioned in the abstract, thus hard to compare. I would suggest sticking to the % metrics also in this instance (or else give kg/km estimates also for the other options).

- In the same sentence, projections for 50% and 100% RES are mixed, which makes it hard to grasp.

Other than that the paper is ready to be published from my point of view.

Ms. Ref. No.: COMMSENG-23-0358C

Title: Solutions for Decarbonizing Urban Bus Transport in non-OECD Countries: A Life Cycle Case Study in Saudi Arabia

Detailed Response to Reviewers

We express our gratitude to the editors and reviewers for dedicating their valuable time to review the latest revision. We are happy with the recommendation to publish our paper following some revisions that have notably improved the paper. The manuscript has been revised to include an expanded introduction clarifying the research background, revising terminology, and modifications to the abstract and content following journal guidelines. We believe these adjustments strengthened the foundation of the manuscript and adequately addressed the reviewers' concerns.

Here are our responses to the reviewers' comments:

Reviewers' comments:

Reviewer #3 (Remarks to the Author):

Thanks for inviting me to review the second revised version of the paper. Again, the authors have gone to great lengths to revise the paper. The request for additional data has been answered to my satisfaction. Moreover, scenarios for 100% renewable electricity-powered BEB (2022 and 2030) and for the hydrogen mix in 2030 have been added. This provides the basis for an unbiased comparison of all options. The additional outcomes show that, in terms of technology, BEB busses run on RES electricity are the most energy efficient solution and achieve the highest life-cycle GHG emission reductions, which confirms previous research results.

I have following further suggestions for minor edits:

- With the addition of the 100%RES scenario, it becomes implicitly clear that the conclusion of blue H2 busses as the most effective option for GHG emission reduction rests on the assumption that blue H2 will become abundantly available in Saudi Arabia, whereas RES electricity will not (at least by 2030). It should, therefore, be made clear why renewable electricity is not viable or feasible for Saudi Arabia. It seems that the barrier is not of technical nature, but lies with the supposed present energy policy strategy of Saudi Arabia to ramp up hydrogen but not renewable electricity. Unless I am mistaken (I screened the paper again in this regard, but no full read), it is

not made sufficiently clear that this strategy exists, if and why Saudi Arabia is not planning to exploit renewable electricity, and how credible and persistent this strategy is out to 2030. A short description of the Saudi Arabian energy strategy, including suitable evidence, should be added (either to the introduction, or in section 2.3 on the fuel cycle) to show of what nature is the barrier for not being able to go for more renewable electricity consumption in transport, and why this is therefore considered “theoretical”, “hypothetical” or not “viable/feasible” in the paper.

We appreciate the reviewer's insightful question regarding the feasibility of renewable electricity for Saudi Arabia and the implications for the adoption of blue hydrogen buses. In our paper, we assumed the continued reliance on blue hydrogen based on the current strategic trajectory of Saudi Arabian energy policy and the relatively slow ramp-up of renewable energy sources.

To address the concerns raised by the reviewer:

1. **Renewable Electricity Viability:** Although the potential of renewable energy in Saudi Arabia is significant, its current utilization remains relatively low due to infrastructure, financial, and policy constraints. These challenges include the high upfront capital costs for renewable infrastructure, limited grid connectivity for distributed renewable generation, and a historical focus on oil and gas as primary energy sources.
2. **Energy Policy Strategy:** Saudi Arabia's Vision 2030 explicitly includes ambitious goals for the expansion of renewable energy sources. However, the implementation pace for renewable electricity generation remains uncertain due to ongoing reliance on fossil fuel exports and a preference for hydrogen as a flexible energy carrier. The strategy aims to diversify the energy mix, but blue hydrogen is seen as a transitional solution due to existing natural gas resources and carbon capture infrastructure (Aramco, 2023; Hasan & Shabaneh, 2021; Mammoser, 2022). Current policy priorities reflect a focus on positioning Saudi Arabia as a leading exporter of blue hydrogen (Aramco, 2023; Hasan & Shabaneh, 2021; Mammoser, 2022). Nevertheless, we have included realistic renewable energy scenarios for the 2030 case which are aligned with the country's stated policy.
3. **Comparison to BEBs and Green Hydrogen FCBs:** A comparison between BEBs powered entirely by renewable energy and FCBs driven by green hydrogen would be more appropriate. However, in our analysis, we excluded green hydrogen-powered FCBs due to the feasibility of green hydrogen and the uncertainty surrounding the deployment of renewable energy in Saudi Arabia.

We clarified these points by providing additional details on Saudi Arabia's energy strategy in the introduction (Manuscript: lines 98-107, page 3).

lines 98-107: “Achieving 100% renewable electricity for BEBs and transitioning to green hydrogen for FCBs present challenges due to uncertainties in renewable electricity implementation and green hydrogen feasibility in the short to medium term [4]. This study evaluates practical decarbonization strategies involving FCBs powered by blue and grey hydrogen, leveraging NG resources and carbon capture infrastructure, as well as grid-electricity BEBs, as practical transitional decarbonization solutions [32]. Additionally, hypothetical scenarios involving fully renewable BEBs and FCBs powered by a projected 2030 hydrogen mix are explored to thoroughly assess their decarbonization potential.”

- In the Conclusions, authors use the terms “feasible bus fleets”, or “viable bus fleets”. I think this formulation is misleading, as on the fleet side, fully electric bus fleets are included in the viable solutions in the paper. It seems therefore that the authors refer to feasible or viable fuel/electricity

production pathways, from which they exclude fully renewable electricity. Here, wording should be adapted to make clearer for the reader that it is not the fleet authors deem not viable (e.g., saying “feasible electricity supply”).

We appreciate the reviewer's observation regarding the terms used in the Conclusions section. We agree that the terms "feasible bus fleets" and "viable bus fleets" may be misleading and could be interpreted as referring to the fleet itself rather than the energy supply pathways.

To address this, we have made the following modifications in the **conclusion, manuscript**:

Clarifications:

Line 424, Page 11: Changed to "When considering feasible fuel supply, our study identified blue FCBs as the leading solution for decarbonizing the urban bus transportation sector within the short-to-medium term."

Line 435-436, Page 11: Changed to "Contrarily, our results demonstrate that when considering a feasible electricity supply, BEBs in the current 2022 scenario do not notably contribute to GHG emissions reduction across the entire bus life-cycle."

Thank you for the helpful suggestion, which has enabled us to improve the precision and clarity of our conclusions.

- Another minor suggested edit regards the abstract, where new text has been added that reads: "With projected technological advancements and 50% renewable energy by 2030, FCBs and BEBs could achieve further carbon reduction by 19.3%-51%, potentially lowering emissions to 0.16 kgCO₂-eq/km for BEBs powered entirely by renewables." The formulation is unclear,

- Does it mean "... by 19.3% and 51% respectively" (i.e., 19.3% for FCB and 51% for BEV)?

- 0.16 kgCO₂/km is the only absolute emission mentioned in the abstract, thus hard to compare. I would suggest sticking to the % metrics also in this instance (or else give kg/km estimates also for the other options).

- In the same sentence, projections for 50% and 100% RES are mixed, which makes it hard to grasp.

We appreciate the reviewer's attention to detail and their suggestion regarding the clarity of the abstract.

The phrasing in the abstract was intended to convey that both grey and blue hydrogen FCBs, along with BEBs, could achieve varying levels of additional carbon reduction by 2030 with projected technological advancements and a 50% renewable energy mix.

To ensure clarity, we have revised the **abstract** to read as follows, in accordance with the guidelines for *Communication Engineering*:

Abstract: "Non-OECD countries like Saudi Arabia face challenges in reducing carbon emissions from urban bus transportation. Herein, we address the gaps in evaluating proton-exchange membrane fuel cell buses and develop a globally relevant life cycle assessment model using Saudi Arabia as a case study. We compare the environmental impact of buses powered by grey and blue hydrogen fuel cells, battery electric motors, and diesel engines in 2022 and 2030 by including the shipping phase, air conditioning load, and refueling infrastructure. The assessment

illustrates fuel cell buses using blue hydrogen can reduce emissions by 53.6% compared to diesel buses, despite a 19.5% increase in energy use from carbon capture and storage systems. Battery electric buses are affected by the energy mix and battery manufacturing, so only cut emissions by 16.9%. Sensitivity analysis shows climate benefits depend on energy sources and efficiencies of carbon capture and hydrogen production. By 2030, grey and blue hydrogen-powered fuel cell buses and battery electric buses are projected to reduce carbon emissions by 19.3%, 33.4%, and 51% respectively, compared to their 2022 levels. Fully renewable-powered battery electric buses potentially achieve up to 89.6% reduction. However, fuel cell buses consistently exhibit lower environmental burdens compared to battery electric buses.”

Other than that the paper is ready to be published from my point of view.

References:

- Aramco. (2023). First accredited low-carbon ammonia shipment for power generation dispatched from Saudi Arabia to Japan. Retrieved from https://www.aramco.com/en/news-media/news/2023/low-carbon-ammonia-shipment?utm_source=&utm_medium=&utm_campaign=&utm_term=&utm_content=&gad_source=1&gclid=CjwKCAjw3NyxBhBmEiwAyofDYUDuO5AUHqdD9WCR2JlqMPj6Nx41I8AP39IeuJWqvDAzvydKGHLrhoC9ckQAvD_BwE
- Hasan, S., & Shabaneh, R. (2021). The Economics and Resource Potential of Hydrogen Production in Saudi Arabia. *KAPSARC: Riyadh, Saudi Arabia*.
- Mammoser, A. P. (2022). Saudi Arabia Bets Big On Blue Hydrogen. Retrieved from <https://www.kapsarc.org/news/saudi-arabia-bets-big-on-blue-hydrogen/>